# The type 1 diabetes gene *TYK2* regulates β-cell development and its responses to interferon-α

Vikash Chandra [1,11] ✉, Hazem Ibrahim [1,11], Clémentine Halliez [2], Rashmi B. Prasad [3,4], Federica Vecchio[2], Om Prakash Dwivedi [4], Jouni Kvist [1], Diego Balboa [1,5], Jonna Saarimäki-Vire [1], Hossam Montaser[1], Tom Barsby[1], Väinö Lithovius[1], Isabella Artner [6], Swetha Gopalakrishnan[7], Leif Groop [4], Roberto Mallone [2,8], Decio L. Eizirik[9] & Timo Otonkoski [1,10] ✉

Type 1 diabetes (T1D) is an autoimmune disease that results in the destruction of insulin producing pancreatic β-cells. One of the genes associated with T1D is *TYK2*, which encodes a Janus kinase with critical roles in type-I interferon (IFN-I) mediated intracellular signalling. To study the role of TYK2 in β-cell development and response to IFNα, we generated *TYK2* knockout human iPSCs and directed them into the pancreatic endocrine lineage. Here we show that loss of TYK2 compromises the emergence of endocrine precursors by regulating KRAS expression, while mature stem cell-islets (SC-islets) function is not affected. In the SC-islets, the loss or inhibition of TYK2 prevents IFNα-induced antigen processing and presentation, including MHC Class I and Class II expression, enhancing their survival against CD8⁺ T-cell cytotoxicity. These results identify an unsuspected role for TYK2 in β-cell development and support TYK2 inhibition in adult β-cells as a potent therapeutic target to halt T1D progression.

Type 1 diabetes (T1D) is a chronic autoimmune disease characterized by pancreatic islet inflammation resulting in eventual specific loss of the insulin-secreting β-cells[1,2]. Genome-wide association and other genetic studies have identified more than 120 non-HLA regions associated with the risk of developing T1D[3]. One such predisposing gene is tyrosine kinase 2 (*TYK2*), a member of the JAK (Janus Kinase) family, which plays a critical role in intracellular signal transducer and activator of transcription (STAT) signalling stimulated by cytokines, including type-I interferons (IFN-I)[4]. IFN-I signalling is involved in the aetiology of T1D through upregulation of MHC Class I expression and antigen presentation, which leads to the targeting of cytotoxic autoimmunity towards β-cells[5,6]. A recent report suggests that rare loss-of-function *TYK2* promoter mutations are associated with increased diabetes susceptibility in the Japanese population[7,8]. Intriguingly, some of the non-synonymous single-nucleotide polymorphisms (SNPs) of *TYK2* (rs34536443 and rs2304256) that induce a partial inhibition of *TYK2* expression are associated with protection against several autoimmune diseases, including T1D and

[1]Stem Cells and Metabolism Research Program, Faculty of Medicine, University of Helsinki, Helsinki 00290, Finland. [2]Université Paris Cité, Institut Cochin, CNRS, INSERM, Paris 75014, France. [3]Department of Clinical Sciences, Diabetes and Endocrinology, Lund University, CRC, Malmö 22100, Sweden. [4]Institute for Molecular Medicine Finland (FIMM), Helsinki University, Helsinki 00290, Finland. [5]Bioinformatics and Genomics Program, Centre for Genomic Regulation, The Barcelona Institute of Science and Technology, Barcelona 08003, Spain. [6]Endocrine Cell Differentiation and Function group, Stem Cell Center, Lund University, Lund 22184, Sweden. [7]Institute of Biotechnology, HiLIFE, University of Helsinki, Helsinki 00790, Finland. [8]Assistance Publique Hôpitaux de Paris, Service de Diabétologie et Immunologie Clinique, Cochin Hospital, Paris 75014, France. [9]ULB Center for Diabètes Research, Université Libre de Bruxelles, Brussels 1070, Belgium. [10]Department of Pediatrics, Helsinki University Hospital, Helsinki 00290, Finland. [11]These authors contributed equally: Vikash Chandra, Hazem Ibrahim. ✉e-mail: vikash.chandra@helsinki.fi; timo.otonkoski@helsinki.fi

rheumatoid arthritis[9,10]. As the majority of the T1D candidate genes are expressed in human islets and regulate their function[2], it is important to decipher their role in pancreatic development, function, and responses to immune challenges, so that stage specific therapeutic interventions can be effectively developed. Human pluripotent stem cells (hPSCs) and their efficient differentiation towards pancreatic stem cell-islets (SC-islets) provide preeminent

tools for studying human pancreatic development and the role for candidate genes acting at the β-cell level[11-15].

We presently studied the expression of T1D candidate genes in the developing endocrine pancreas and identified *TYK2* as one of the upregulated genes. The effects of TYK2 perturbation are studied by single-cell transcriptomics in the SC-islets throughout their development. Surprisingly, specific *TYK2* knockout (KO) stem cell lines show

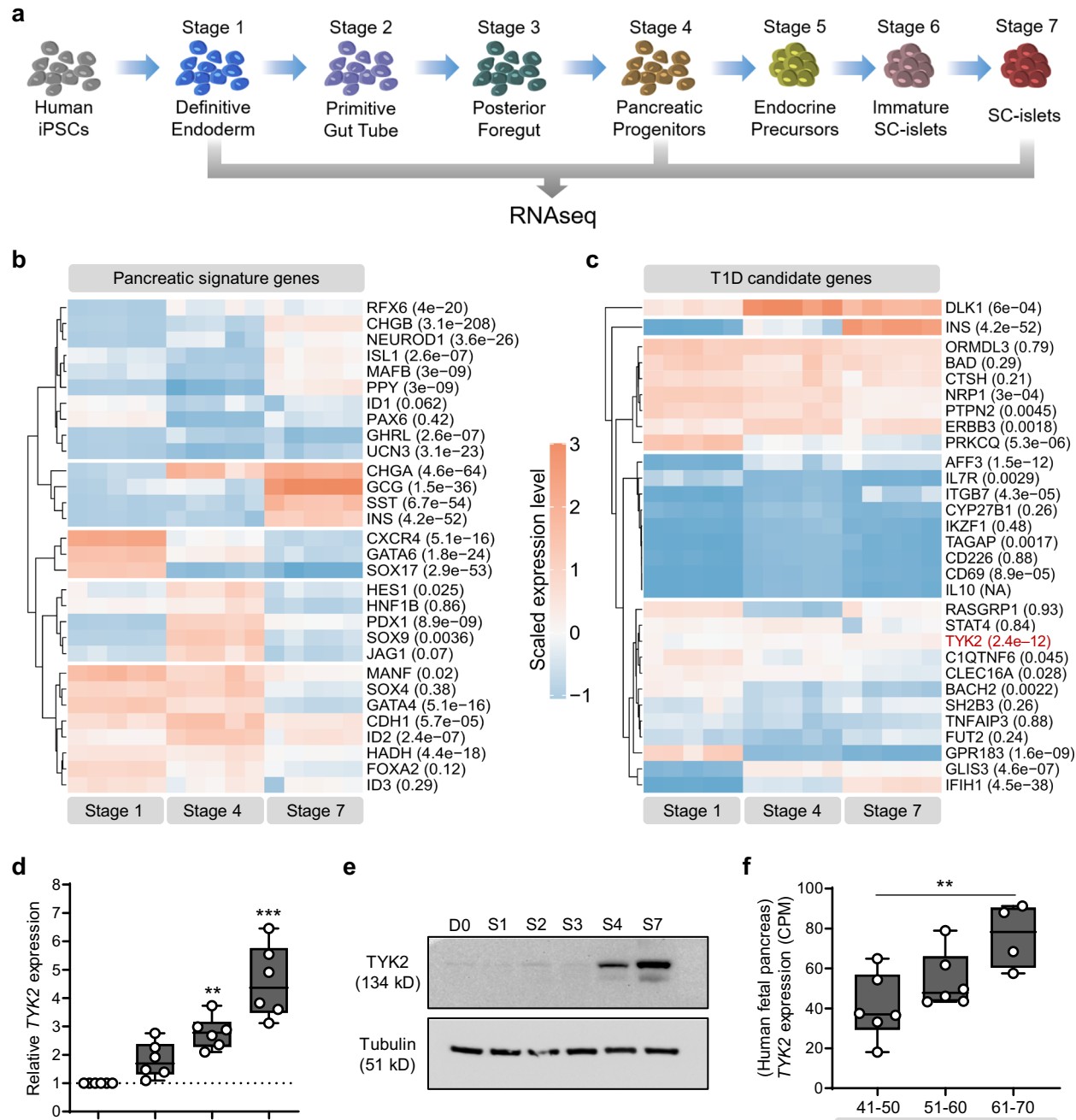

**Fig. 1 | Expression pattern of *TYK2*, a T1D candidate gene, during human pancreatic development. a** Schematic representation of the pancreatic differentiation protocol. The figure was partly generated using Servier Medical Art. **b** Heatmap of pancreatic lineage genes from deep RNAseq analysis at Stage (S) 1 (definitive endoderm), S4 (pancreatic-progenitors), and S7 (SC-islets). **c** Similar heatmap for known T1D candidate genes. Each gene is shown with a multiple testing corrected *p* value generated for the longitudinal differential expression of the gene during differentiation (*n* = 5). **d** Relative expression of *TYK2* during

pancreatic differentiation shown by qRT-PCR (*n* = 6) and **e** by immunoblot analysis for TYK2 protein. Tubulin was used as a loading control (*n* = 3). **f** Expression pattern of *TYK2* in human fetal pancreas samples at 40 to 70 days post conception (*n* = 16). For **d** and **f**, significance was determined using one-way ANOVA with Tukey's multiple comparison test, box and whiskers plots showing the median with whiskers extending from minimum to maximum values. *p < 0.05; **p < 0.01; ***p < 0.001. Source data are provided as a Source Data file.

compromised endocrine precursor formation, whereas TYK2 inhibition in differentiated islet cells completely blocks the IFNα responses. Based on our results, TYK2 inhibition of mature islet cells is a potential preventive therapy for T1D.

## Results

### Expression of T1D candidate genes during pancreatic differentiation

To identify candidate genes of pancreatic development that are also associated with the development of T1D in humans, we performed deep RNA sequencing (~200 million reads per sample) at specific stages of human induced pluripotent stem cells (hiPSCs) pancreatic differentiation (Fig. 1a). Our differentiation strategy robustly phenocopied human pancreatic development as revealed by the expression pattern of key genes of pancreatic developmental hierarchy (Fig. 1b). Screening of stage-specific transcriptome revealed differential expression of several T1D candidate genes, e.g., *TYK2*, *DLK1*, and *IFIH1* were strongly upregulated while *PRKCQ*, *BACH2*, and *GPR183* were downregulated as differentiation progressed towards endocrine lineage (Fig. 1c). The significant increase of TYK2 expression (*p*adj = 2.4E-12) was further confirmed at transcript and protein level in independent experiments (Fig. 1d, e). In agreement with this observation, RNAseq data on human fetal pancreas of mid-to-late 1st trimester (*n* = 16) showed a strong upregulation of *TYK2* transcription with increasing gestational age (Fig. 1f).

### Loss or inhibition of TYK2 results in defective EP formation

Next, we employed CRISPR-Cas9 to generate *TYK2* KO hiPSC lines (HEL46.11)[12]. Two KO clones, selected by the loss of TYK2 protein expression, were used in this study and a non-edited negative clone served as a wildtype (WT) control (Supplementary Fig. 1a–c). Sanger sequencing confirmed the precise deletion of the targeted 277 bp sequence of the ATG-start codon-containing exon 3 (Supplementary Fig. 1d) with no evidence of CRISPR-induced off-target indels. Bulk RNAseq based e-Karyotyping[16] and immunocytochemical analysis of WT and *TYK2* KO hiPSC lines confirmed the absence of genomic aberrations and showed similar expression levels for key proliferation and pluripotency markers (Supplementary Fig. 1e–h).

TYK2 is known to associate with IFN-I receptor (IFNAR1) but not IFN-II receptor and activate downstream STAT signalling[17]. Stimulation with IFNα activated STAT1 and STAT2 in the WT but not in the KO (Fig. 2a). In contrast, IFNγ treatment activated STAT1 in both WT and *TYK2* KO cells, further confirming the specific inhibition of IFNAR1 signalling in *TYK2* KO cells.

To study the role of TYK2 in pancreatic development, we performed pancreatic differentiation using a seven-stage protocol[12,15]. The WT and *TYK2* KO lines showed comparable differentiation capacity until the pancreatic progenitors (PP) stage (S) 4, as demonstrated by similar proportions of cells expressing CXCR4 at S1 (definitive endoderm) and co-expressing PDX1 and NKX6-1 at S4 (Supplementary Fig. 2a, b). In addition, the expression levels of *SOX9*, *FOXA2*, *PTF1A*, and *PDX1* were similar at S4. However, NKX6-1 expression was significantly reduced in the KO compared to their WT counterpart (Supplementary Fig. 2c).

Endocrine precursors (EP) induced at S5 of the differentiation protocol. EP formation was significantly compromised in *TYK2* KO, as shown by decreased expression of transcripts for *NKX6-1* (25 ± 7% reduction), *NEUROG3* (56 ± 9%), and *NKX2-2* (58 ± 7%; Fig. 2b). The number of NEUROG3⁺ cells was also significantly reduced (46.3% reduction, *p* = 0.0002) as evaluated by immunohistochemistry (Fig. 2c). The proportion of PDX1⁺NKX6-1⁺ cells was significantly reduced as evaluated by flow cytometry (19 ± 4% reduction, *p* = 0.004) (Fig. 2d).

S6 is marked by a substantial increase of immature endocrine cell numbers. As a consequence of impaired EP formation, we also observed a global decrease in the expression of the islet hormone

transcripts *INS* (59 ± 8% reduction), *GCG* (70 ± 7%) and *SST* (54 ± 10%) at S6 (Fig. 2e). This observation was further confirmed by flow cytometry for INS⁺NKX6-1⁺ cells (29.8 ± 2% reduction) and normalized total insulin content (51.4 ± 8% reduction) (Fig. 2f–h). Further characterization at S7 confirmed ≈32% reduced number of monohormonal INS⁺ cells in *TYK2* KO SC-islets (Fig. 2i, j), while the number of single-positive GCG⁺, SST⁺, and Ki-67⁺ cells were comparable to WT (Supplementary Fig. 3a–d). SC-islet differentiation has been reported to generate non-pancreatic enterochromaffin like cells (SC-EC) which follow the developmental lineage of β-cells[18]. Interestingly, we also detected a significant ≈31% reduction in the SC-EC (SLC18A1⁺) cells in *TYK2*-KO S7 SC-islets (Fig. 2j, k).

Static and dynamic insulin secretion was studied after 3 weeks of S7 maturation. In agreement with the reduced number of mono-hormonal INS⁺ cells, we detected lower secreted insulin in *TYK2* KO SC-islets (Fig. 2l). However, the functionality of the KO β-cells was normal, evidenced by an intact insulin stimulatory index to high glucose, to the glucagon-like peptide-1 (GLP1) analogue exendin-4 and to K⁺-induced membrane depolarization (Fig. 2m). To further investigate the in vivo functional potential of the S7 mature SC-islets, we implanted equal numbers of WT and *TYK2* KO S7 SC-islets under the kidney capsule of non-diabetic NOD-SCID-Gamma mice (Supplementary Fig. 4a). Blood glucose and circulating human C-peptide levels were comparable between WT or *TYK2* KO S7 implanted mice measured until 2 months post-implantation (Supplementary Fig. 4b, c). The proportion of INS⁺ and GCG⁺ cell populations measured as ratio of mono-hormonal INS⁺ or GCG⁺ cells to total INS⁺ and GCG⁺ cells in WT and *TYK2* KO 2-month grafted tissue sections were unchanged (Supplementary Fig. 4d, e). These results indicate that the complete loss of TYK2 impaired S5 EP formation thus reduced the number of endocrine β-cells but did not affect the subsequent in vitro or in vivo maturation and function of β-cells.

Notably, the RNA-seq data from 16 human fetal pancreata also revealed a significant positive correlation between *TYK2* and *NEUROG3* (*r* = 0.63; *p* = 0.026; Fig. 2n). Likewise, the adult islet RNA-seq dataset[19] (*n* = 191) showed a strong positive correlation between *TYK2* and *INS* expression (*r* = 0.77; *p* = 4.4E-39; Fig. 2o), suggesting a regulatory role of TYK2 in the pancreatic endocrine lineage.

We next validated these findings by using a selective and potent allosteric TYK2 inhibitor (TYK2i) BMS-986165[20] during S3 to S5 differentiation. Upon IFNα stimulation of S5 WT cells, STAT1 and STAT2 activation were abolished by TYK2i (Fig. 2p). We then investigated whether TYK2i could recapitulate the *TYK2* KO reduced EP formation phenotype. Following TYK2i treatment at the end of S5, we also observed a similarly reduced expression of the transcripts *PDX1* (by 30 ± 3%), *NEUROG3* (by 33 ± 4%), *NKX6-1* (by 28 ± 6%), and *NKX2-2* (by 26 ± 8%) with no change in the TYK2i treated KO samples (Fig. 2q). Similarly, we replicated the above findings of TYK2 inhibition on the H1 human embryonic stem cells (hESCs) (Supplementary Fig. 5a–e). Collectively, these data confirm that TYK2 regulates the formation of pancreatic EP.

### TYK2 negatively regulates KRAS expression

To understand the mechanism by which loss or inhibition of TYK2 compromises EP formation during pancreatic lineage differentiation, a whole transcriptome analysis was performed at the stem cell level (hiPSCs), S4 (PP), S5 (EP), and S6 (immature SC-islets; Fig. 3a). Principal component analysis (PCA) of respective transcriptomes for WT and KO genotype clustered together in order of developmental stage, but a significant difference was observed at S5 (Fig. 3b), with 319 upregulated and 412 downregulated genes (Fig. 3c). In line with the findings described above, the key pancreatic transcription factors (TFs) *NEUROG3* (*p*adj = 7.4E-6) and *NKX2-2* (*p*adj = 4.1E-3) were significantly downregulated at S5 (Fig. 3d, e). Reactome enrichment analysis confirmed the downregulation of gene sets associated with "Regulation of

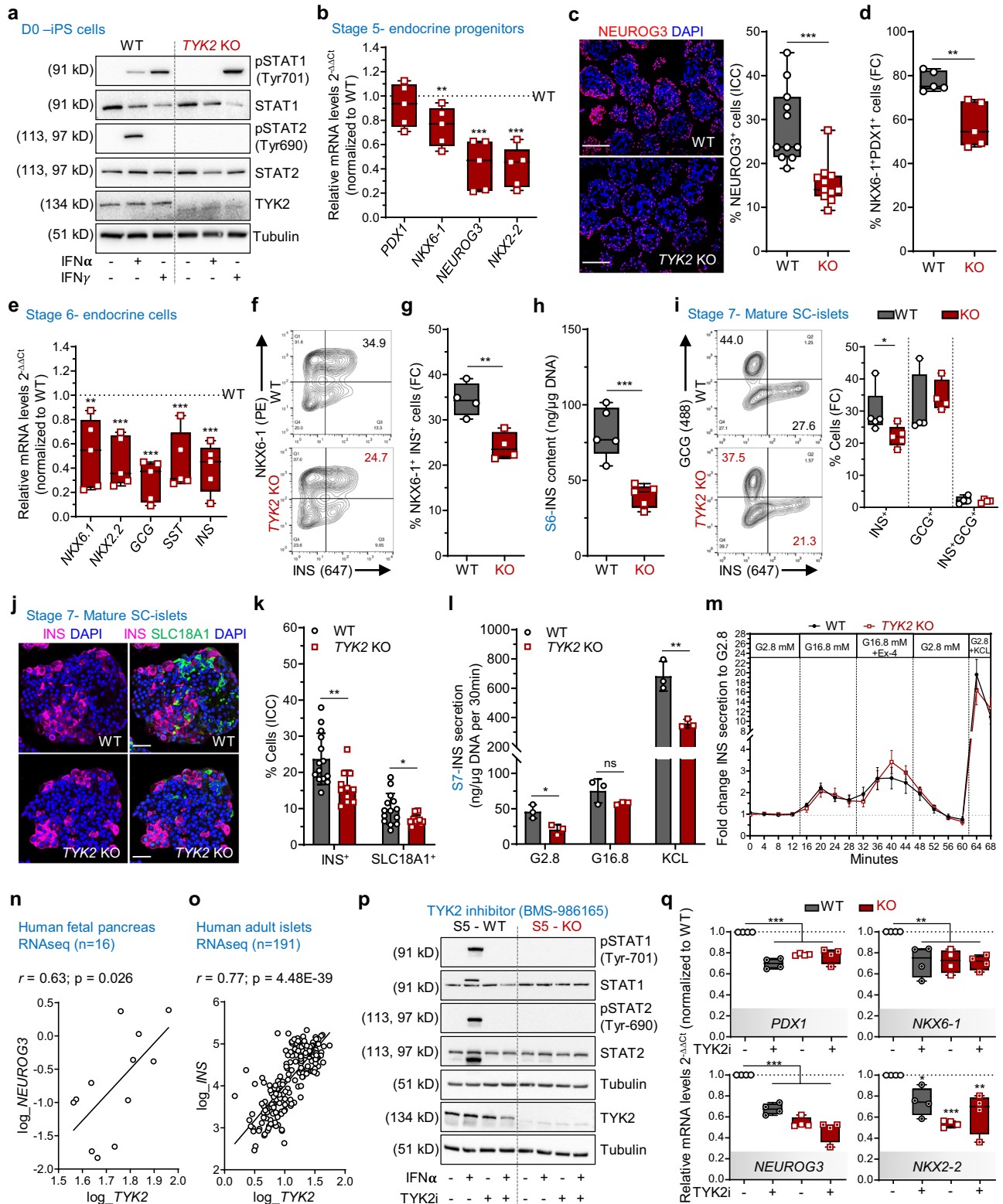

β-cell Development (e.g., *INSM1, NEUROD1, ONECUT1, PAX4, PAX6*)" and "Gene Expression in β-cells", while "Signalling by Receptor Tyrosine Kinases (e.g., *DUSP6, KRAS,* SPP1, *LAMB2, COL2A1*)" was unexpectedly upregulated (Fig. 3f).

There were only 16 consistently upregulated and three down-regulated genes throughout all stages of differentiation. Strikingly, we observed that the most significantly upregulated gene at all stages was *KRAS* (Fig. 3g). This observation was confirmed at gene (qRT-PCR) and protein (WB) levels (Fig. 3h, i). Similarly, we observed a significant

increase of KRAS protein upon TYK2i treatment in WT, but not KO cells, indicating a negative correlation between TYK2 and KRAS (Fig. 3j, k). The RNAseq data from human fetal pancreata ($r = -0.51$; $p = 0.042$) and adult islets ($r = -0.56$; $p = 2.06E-17$) also showed a significant negative correlation between *TYK2* and *KRAS* expression levels (Fig. 3l, m). Moreover, a strong negative correlation between *KRAS* and *INS* ($r = -0.61$; $p = 1E-4$) was also found in the islet dataset (Fig. 3n). Next, we transiently overexpressed TYK2 (pCMV-TYK2) during S5 in WT and TYK2 KO cells (Fig. 3o) and confirmed the upregulation of *TYK2*

**Fig. 2 | Loss of TYK2 is associated with defective formation of endocrine precursors. a** Immunoblots for IFNα or IFNγ treated WT and *TYK2* KO hiPSCs (*n* = 3). **b** Stage (S) 5 relative transcript levels of *PDX1*, *NKX6-1*, *NEUROG3*, and *NKX2-2* (*n* = 5). **c** Immunohistochemistry for NEUROG3 (scale bar = 150 μm) and quantification of 11 images from 3 experiments (*n* = 11). **d** S5 flow cytometry for PDX1⁺ NKX6-1⁺ cells (*n* = 5). **e** S6 relative transcript levels of *NKX6-1*, *NKX2-2*, *GCG*, *SST*, and *INS* (*n* = 5). **f** S6 flow cytometry for NKX6-1⁺ and INS⁺ cells. **g** Quantification (*n* = 4). **h** S6 total insulin content normalized to the DNA content (*n* = 5). **i** S7 flow cytometry for INS⁺ and GCG⁺ cells (*n* = 4-5). **j** Immunohistochemistry for INS and SLC18A1 in S7 SC-islets (scale bar = 50 μm) and **k** quantification from 2 experiments with 5-7 images each (*n* = 11-14). The number of cells quantified for WT and KO were 10516 and 11990 cells, respectively. **l** S7 static insulin secretion, with low (2.8 mM), high (16.8 mM) glucose and with 30 mM KCl, normalized to the DNA content (*n* = 3). Bar plots in **k** and **l** are means ± S.D. **m** S7 dynamic insulin secretion with 16.8 mM glucose, 50 ng/ml exendin-4 (Ex4), and 30 mM KCl, normalized to the basal secretion. Unpaired *t*-test with Holm-Šídák multiple comparison, data are means ± SEM (*n* = 5). **n** Correlation of *TYK2* and *NEUROG3* expression in human fetal pancreas (*n* = 16). **o** Correlation of *TYK2* and *INS* expression in adult human islets (*n* = 191). Pearson's *r* correlation test after log normalization of counts. **p** Immunoblots for WT and KO treated with TYK2i during S3–S5 analysed for STATs phosphorylation in presence of IFNα (*n* = 3) and **q** relative transcript levels of *PDX1*, *NKX6-1*, *NEUROG3*, and *NKX2-2*. Ordinary one-way ANOVA with Tukey's multiple comparison (*n* = 4). Two-tailed unpaired *t*-test (**b, c, d, e, g, h, i, k, l**). Box and whiskers plots showing median with min to max whiskers. **p* < 0.05; ***p* < 0.01; ****p* < 0.001; ns—non-significant. Source data are provided as a Source Data file.

transcripts and protein (Fig. 3p, q). Importantly, restoration of TYK2 protein in KO cells rescued the *KRAS* upregulation whereas the levels were unchanged in WT cells (Fig. 3r). However, the effects on *NEUROG3* expression were not significant (Fig. 3s).

Collectively, these findings are in agreement with earlier studies showing that high KRAS antagonizes endocrine neogenesis in the developing pancreas through inhibition of NEUROG3 expression[21] and suggest that TYK2 loss or inhibition compromises EP formation by upregulating KRAS.

### Increased *KRAS* perturbs *NEUROG3* induction

To understand the impact of *TYK2* KO at the level of specific pancreatic cell subtypes, we investigated the *TYK2* KO phenotype via single-cell (sc) RNA-seq (Fig. 4a). We generated a dataset of 12545 WT and 13856 KO cells for EP (S5) and SC-islets (S6) and analysed it using the Seurat pipeline[22]. After quality filtering, we annotated the clusters according to the expression of known cell-type specific pancreatic markers (Fig. 4b and Supplementary Fig. 6a). Next, we classified the clusters broadly in four groups: PP (*SOX9⁺⁺ONECUT1⁺⁺TCF7L2⁺⁺*), EP (*NEUROG3⁺⁺*), α-like cells (*GCG⁺⁺ARX⁺⁺*), and β-like cells (*INS⁺⁺NKX6-1⁺⁺*). We observed a marked difference in the normalized distribution of cells in the different clusters emerging from S5 and S6 WT and TYK2 KO cells (Supplementary Fig. 6b–e). WT S6 contained 31.1% β-like cells compared to 15% in KO, whereas S6 KO had still 76.4% PP-like cells compared to 53.1% in WT (Fig. 4c–f). GSEA (Reactome) analysis for PP and EP clusters at S5 confirmed the downregulation of the gene set for "Regulation of β-cell Development", whereas gene sets for "M Phase" and "Cell Cycle" were upregulated in KO (Fig. 4g). Furthermore, we confirmed the reduced expression of *NEUROG3* and its downstream target *NKX2-2* in KO EP cells, in line with our previous observations (Fig. 4h).

scRNA-seq revealed that the expression of *KRAS* was highest in the PP- and EP-like *TYK2* KO clusters (Fig. 4i). *KRAS* has been shown to accelerate the G₁/S transition of the cell cycle, leading to the shortening of G₁ length[23]. Lengthening of the G₁ phase in the PP cells is key to the proper augmentation of *NEUROG3* and its downstream targets[24]. We observed a higher expression of cell division protein kinase 2 (*CDK2*) and *CDK4* in the *TYK2* KO S5 EP cells (Fig. 4j, k). We then used the expression level of cell cycle markers to estimate the fraction of dividing cells in each cluster with Seurat (CellCycleScoring)[22]. Notably, in WT samples 53.6% (of S5 PP) and 37.1% (of S5 EP) cells were in G₁ phase compared to 42% and 24.5% in KO, respectively (Fig. 4l).

Collectively, these data indicate that elevated KRAS drives the G₁/S transition faster in the *TYK2* KO PP and EP cells, resulting in perturbation of G₁ phase that interferes with the proper induction of *NEUROG3* expression. Accordingly, we observed 39% more NKX6-1⁺EdU⁺ double-positive cells with flow cytometry and a significantly higher Ki-67 nuclear staining in the KO S5 clusters (Fig. 4m–p).

### TYK2 regulates the IFNα response in SC-islets

Several studies reported the prominent role of IFNα in the induction of autoimmunity in T1D[25]. Since TYK2 is important for the IFN-I signalling

mediated through IFNAR1, we took advantage of our *TYK2* KO model to understand its role during the dialogue between IFNα and the developing pancreatic islet cells. The phosphorylation of STAT1/2 was completely abrogated in the S6 KO SC-islets in response to IFNα but not IFNγ, whereas STAT3 responses were only partially affected (Fig. 5a).

Next, we performed scRNA-seq on S6 WT (5202 cells) and KO (4281 cells) islet cells treated with IFNα. Notably, the global transcriptomic changes following IFNα treatment in WT individual islet cells were in good agreement with our previously reported human islet dataset under similar treatment[5] (Supplementary Fig. 7a). IFNα induced the upregulation of previously described IFN-stimulated genes (ISGs) in the WT cells. However, this response was absent in *TYK2* KO cells, shown by a heatmap for the top 55 (46 up and 9 down) differentially expressed genes in all clusters (Fig. 5b and Supplementary Fig. 7b). The IFNα response was similarly completely inhibited in both β- and α-like cells (Fig. 5c). Furthermore, genes important for antiviral responses to IFNs like *STAT1* and antiviral MX dynamin-like GTPase 1 (*MX1*) were strongly induced in WT but remained unchanged in the *TYK2* KO cells (Supplementary Fig. 8a). Given that MHC Class I upregulation is one of the most important hallmarks in diabetic β-cells[26], we observed that upon IFNα treatment, *HLA-A*, *HLA-B*, *HLA-C*, and *HLA-E* genes were strongly upregulated in WT cells, but unchanged in *TYK2* KO cells (Supplementary Fig. 8b). In agreement with the experimental data, a strong positive correlation between *TYK2* and *HLA-A* (*r* = 0.52; *p* = 6.53E-15); *HLA-B* (*r* = 0.42; *p* = 1.06E-9); *HLA-C* (*r* = 0.60; *p* = 1.55E-20); and *HLA-E* (*r* = 0.51; *p* = 3.65E-14) was observed in the human islet RNA-seq dataset (*n* = 191)[19] (Supplementary Fig. 8c).

In line with the other findings described above, GSEA (KEGG) analysis highlighted lower expression of antigen processing and presentation gene sets in *TYK2* KO β- and α-like cells following IFNα compared to WT (Fig. 5d). We then analysed the expression of genes associated with antigen processing following IFNα treatment. First, we observed that constitutive proteasome genes (*PSMB5*, *PSMB6*, and *PSMB7*) remained unchanged in both WT and KO while IFNα upregulated genes encoding the immunoproteasome (*PSMB8, PSMB9,* and *PSMB10*) only in the WT. Second, transporters associated with antigen processing (*TAP1, TAP2,* and *TAPBP*) and MHC Class I (*HLA-A, HLA-B, and HLA-C*) were strongly augmented in the WT β-like cells but remained unchanged in the KO[27] (Fig. 5e, f).

Collectively, these data demonstrate that inhibition of TYK2 signalling in β- and α-like cells inhibited both antigen processing and presentation following IFNα treatment. Because of the partially compromised endocrine differentiation of *TYK2* KO SC-islets, these findings could also be related with a lower expression of T1D associated β-cell autoantigens, notably insulin. We therefore compared the expression of previously described T1D autoantigens between WT and *TYK2* KO β-like cells[28]. A significantly lower expression was found only for *GAD1* (in the presence of IFNα) and *GAD2* in *TYK2* KO β-like cells (Supplementary Fig. 8d, e). We next studied the expression of MHC Class II genes, which harbour T1D-associated polymorphisms[29] and are overexpressed in T1D donor islet β-cells[30]. Importantly, we observed a

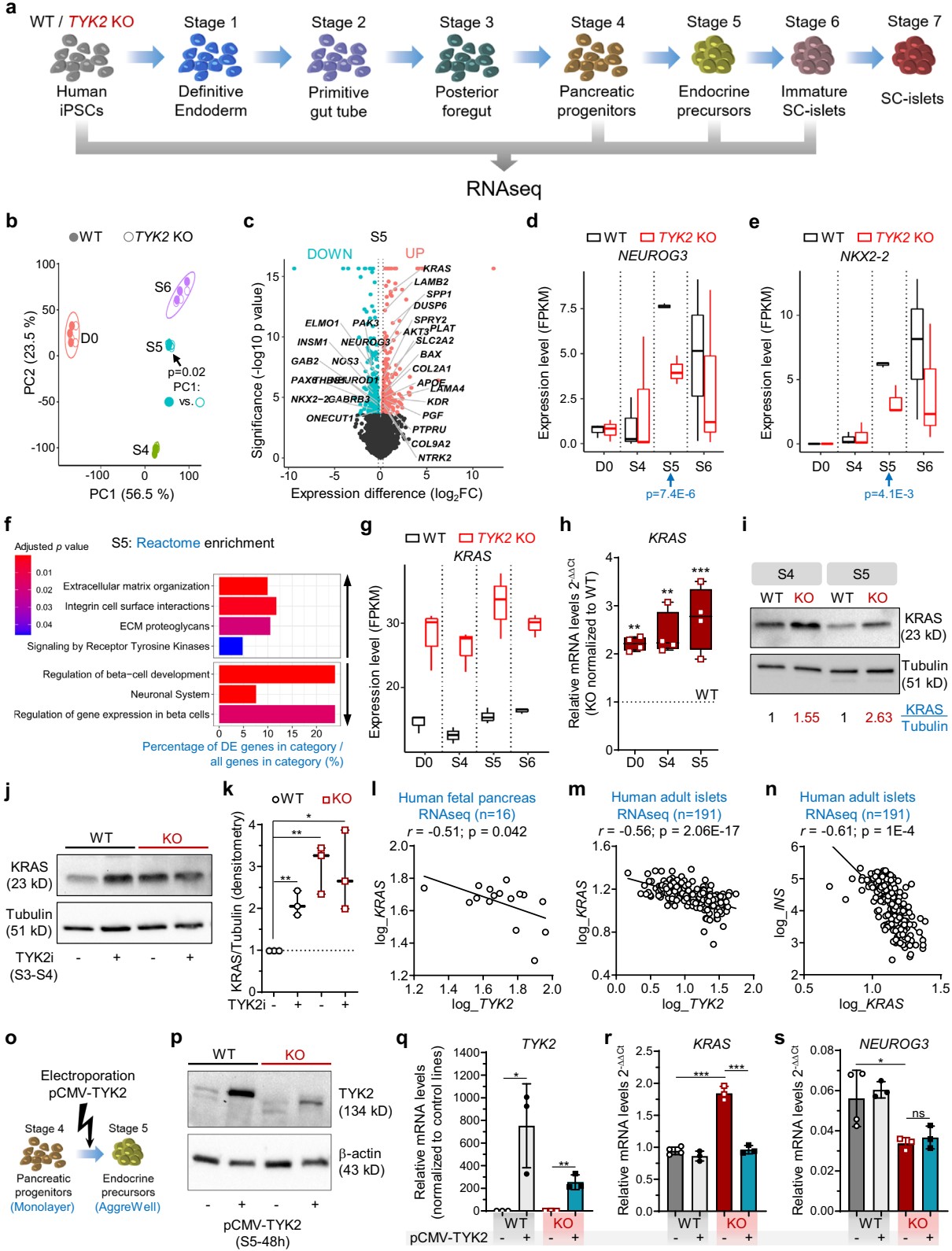

TYK2 dependent upregulation of MHC Class II genes *HLA-DQB1* and *HLA-DRB1* but not *HLA-DPB1* upon IFNα treatment in the S7- SC-islets (Supplementary Fig. 9a–h).

## TYK2 modulates T-cell-mediated cytotoxicity in SC-islets

Next, we examined the expression of MHC Class I protein on the surface of maturing SC-islets following IFNα treatment. We observed a strong upregulation on the WT SC-islets, whereas this response was absent in *TYK2* KO or upon TYK2i co-treatment (Fig. 6a, b). Similarly, flow cytometry confirmed the overall 83.1% MHC Class I increase in WT SC-islets compared to 0.69% in *TYK2* KO (Fig. 6c–e). Notably, *TYK2* KO SC-islets also had 26.9% INS+ cells compared to 41.8% in WT SC-islets.

To investigate whether inhibiting IFNα-induced MHC Class I upregulation by TYK2i could reduce the vulnerability of WT SC-islets

**Fig. 3 | *TYK2* negatively regulates *KRAS* expression. a** Schematic for bulk RNAseq experiments for WT and *TYK2* KO cells (*n* = 3). The figure was generated using Servier Medical Art. **b** Principal component (PC) analysis of the bulk RNA-seq samples. Filled circles, WT cells; empty circles, KO cells. *p*-value (linear regression comparison) PC1-genotype of Stage (S) 5 is indicated. **c** Volcano plot for S5 differentially expressed genes for WT and KO. Significantly downregulated genes, iris blue; upregulated genes, soft red. Selected genes of interest are highlighted. **d**, **e** FPKM values for *NEUROG3* and *NKX2-2* expression at different differentiation stages. *p*-value (DESeq2) is indicated for S5. **f** Selected up- and down- regulated Reactome enrichment pathways in S5 KO cells compared to WT. **g** FPKM values for *KRAS* expression. **h** Relative transcript levels of *KRAS* with qRT-PCR in hiPSCs, S4 and S5 cells. **i** Immunoblot for KRAS at S4 and S5. Normalization with tubulin densitometric values are indicated in the panel (*n* = 3). **j** Immunoblot for KRAS at S4 in WT and KO cells following TYK2i treatment during S3–S4. **k** Densitometric

analysis of panel **j** (*n* = 3). **l** Correlation of *TYK2* and *NEUROG3* expression in human fetal pancreas (*n* = 16). Pearson's correlation after log normalization of counts. **m** Correlation of *KRAS* and *TYK2* expression; and **n** *KRAS* and *INS* expression in human islets (*n* = 191). Pearson's correlations *r* and significance levels *p* are indicated in the panels. **o** Schematic of TYK2 overexpression experiments. **p** Immunoblot for TYK2 48 h post pCMV-TYK2 electroporation during S5. β-actin used as a loading control (*n* = 2). qRT-PCR relative transcript levels 24 h post pCMV-TYK2 overexpression during S5 for **q** *TYK2*, **r** *KRAS*, and **s** *NEUROG3* (*n* = 3). Two-tailed unpaired *t*-tests were applied. *$p < 0.05$; **$p < 0.01$; ***$p < 0.001$; ns−non-significant. The boxplots in **d**, **e**, **g**, **h** and **k** showing the median with lower and upper hinges corresponding to the first and third quartiles (the 25th and 75th percentiles) with min to max whiskers (*n* = 3), except for **h**, (*n* = 4). Source data are provided as a Source Data file.

to T-cell-mediated cytotoxicity, we designed a T-cell co-culture experiment (Fig. 7a). To exclude the potential confounding effect of β-cell antigen downregulation upon TYK2 loss, we used a flu peptide-reactive CD8⁺ T-cell line. Variable T-cell numbers were cultured for 6 h with a fixed mixture of peptide-pulsed, CFSE-labelled SC-islets and unpulsed, CTV-labelled SC-islets, which were pre-treated with IFNα alone or in combination with TYK2i or left untreated. The ratio of surviving SC-islets thus provided a readout of peptide-specific lysis (Fig. 7b). IFNα treatment led to a significant increase in SC-islet lysis, which was inhibited in the presence of TYK2i (Fig. 7c). These cytotoxic outcomes were paralleled by MHC Class I expression (Fig. 7d, e). With increasing T-cell numbers, SC-islets that survived cytotoxicity were those with lower MHC Class I expression (Fig. 7e). Although the inhibition of MHC Class I upregulation by TYK2i was complete, cytotoxicity was only partially inhibited. In line with the known effect of IFNα on programmed death ligand-1 (PD-L1) upregulation[31], TYK2i treatment also partially inhibited PD-L1 upregulation (Fig. 7d, f). PD-L1^high SC-islets were those surviving cytotoxicity with increasing T-cell numbers up to 5:1 effector-to-target ratios (Fig. 7f). The concomitant inhibitory effect of TYK2i on MHC Class I and PD-L1 upregulation may explain the incomplete inhibition of T-cell mediated cytotoxicity.

Collectively, these results demonstrate that TYK2 inhibition prevents IFNα-induced MHC Class I upregulation on SC-islets and, despite some concomitant inhibition of PD-L1 upregulation, significantly decreases the ensuing T-cell-mediated cytotoxicity.

## Analysis of *TYK2* SNPs (rs34536443 and rs2304256) in the Finn-Gen cohort

Loss-of-function SNPs of *TYK2* (e.g., rs34536443, rs2304256) have been reported to be protective against several autoimmune diseases including T1D[9,32]. The TYK2i BMS-986165 used in the present study mimics the effect of the rs34536443 (TYK2^P1104A) coding variant[20]. We replicated the phenome-wide association analysis for these SNP in the R7 dataset of FinnGen project (r7.finngen.fi), which compiled 3095 clinical endpoints obtained from electronic health record data of 309,154 Finnish individuals (Supplementary Table 1). We found that rs34536443 and rs2304256 provided protection from several autoimmune/auto-inflammatory diseases in the Finnish population, including T1D (8671 cases and 255,466 controls; Supplementary Fig. 10a, b).

## Discussion

Here we demonstrate a previously unknown role of the T1D candidate gene *TYK2* in pancreatic endocrine cell development. In the absence of TYK2, KRAS expression was upregulated, which resulted in improper induction of the pro-endocrine transcription factor NEUROG3. Although the mature *TYK2* KO SC-islets had lower numbers of β-cells, they had intact insulin secretory machinery, shown by the normal insulin stimulatory index response (Fig. 2i–m and Supplementary Fig. 4b, c). Importantly, TYK2 was found to be essential for the IFN-I

responsiveness of SC-islets, supporting the therapeutic rationale of TYK2 inhibition to halt T1D progression.

Directed differentiation of hPSCs into pancreatic islet cells provides a controlled experimental system to study the role of diabetes-associated genes in pancreas development[11,33]. We and others have previously shown that STAT3 activation regulates the NEUROG3 mediated pancreatic β-cell differentiation[12,34]. In the present study, we show that TYK2 is a major STAT activator that also has a regulatory role on NEUROG3 expression in pancreatic progenitors. Interestingly, proinflammatory cytokines (IL1β, TNFα, and IFNγ) induce endocrine differentiation in pancreatic ductal cells through STAT3 dependent NEUROG3 expression[35].

Our results revealed an unexpected and previously unknown negative regulation between *TYK2* and *KRAS* (Fig. 3). Regulation of KRAS expression by TYK2 in KO cells is noteworthy, since JAK/STAT signalling is known to be important for receptor tyrosine kinase (RTK) and mitogen-activated protein kinase (MAPK) signalling[36]. In agreement with this, Reactome enrichment analysis of S5 bulk RNA-seq revealed a significant upregulation of RTK pathways in the *TYK2* KO cells. It has been shown that KRAS accelerates the $G_1/S$ progression and lengthening of the $G_1$ phase is a prerequisite for the complete induction of NEUROG3 expression before cell-cycle exit of the endocrine progenitors[24]. Thus, an elevated KRAS level in *TYK2* KO PP and EP cells accelerates their cell cycle, resulting in compromised NEUROG3 augmentation (Fig. 4l–n). Importantly, KRAS has been shown to not possess oncogenic functions in pancreatic endocrine cells and activating KRAS mutations were never found in islet cell tumors (Supplementary Fig. 3c, d)[21,37]. In agreement with this, we observed no differences in the cell cycle dynamics of *TYK2* KO β-like cells after S5 of differentiation. Taken together, we show that regulation of the JAK/STAT pathway in pancreatic progenitors by TYK2 is essential for the cell-cycle control required for the progression of endocrine differentiation, but TYK2 is not involved in the regulation of proliferation in the mature endocrine cells.

Conspicuously, rare loss-of-function *TYK2* promoter mutations (Clin Var, 440728) in the Japanese population predispose to autoantibody-negative T1D and T2D[8]. This association has been thought to be linked with increased susceptibility to viral infection[38]. However, based on our results, the partial loss of TYK2 expression could lead to a lower β-cell mass as a contributing factor to the increased diabetes risk.

Notably, a germline loss-of-function *TYK2* mutation (rs34536443; TYK2^P1104A) has been reported and hereby confirmed in the Finnish population to be protective against several autoimmune diseases including T1D[9,32] (Supplementary Fig. 10). This highly protective effect against autoimmunity is associated with the dampening of IFN-I, IL12 and IL23 signalling[32]. Of note, IFN-I responses are important contributors to T1D etiology and inhibition of these responses in early disease has been suggested as a potential intervention[6,39]. *TYK2*-silenced adult β-cells (i.e., a 50% inhibition induced by siRNA or TYK2i)

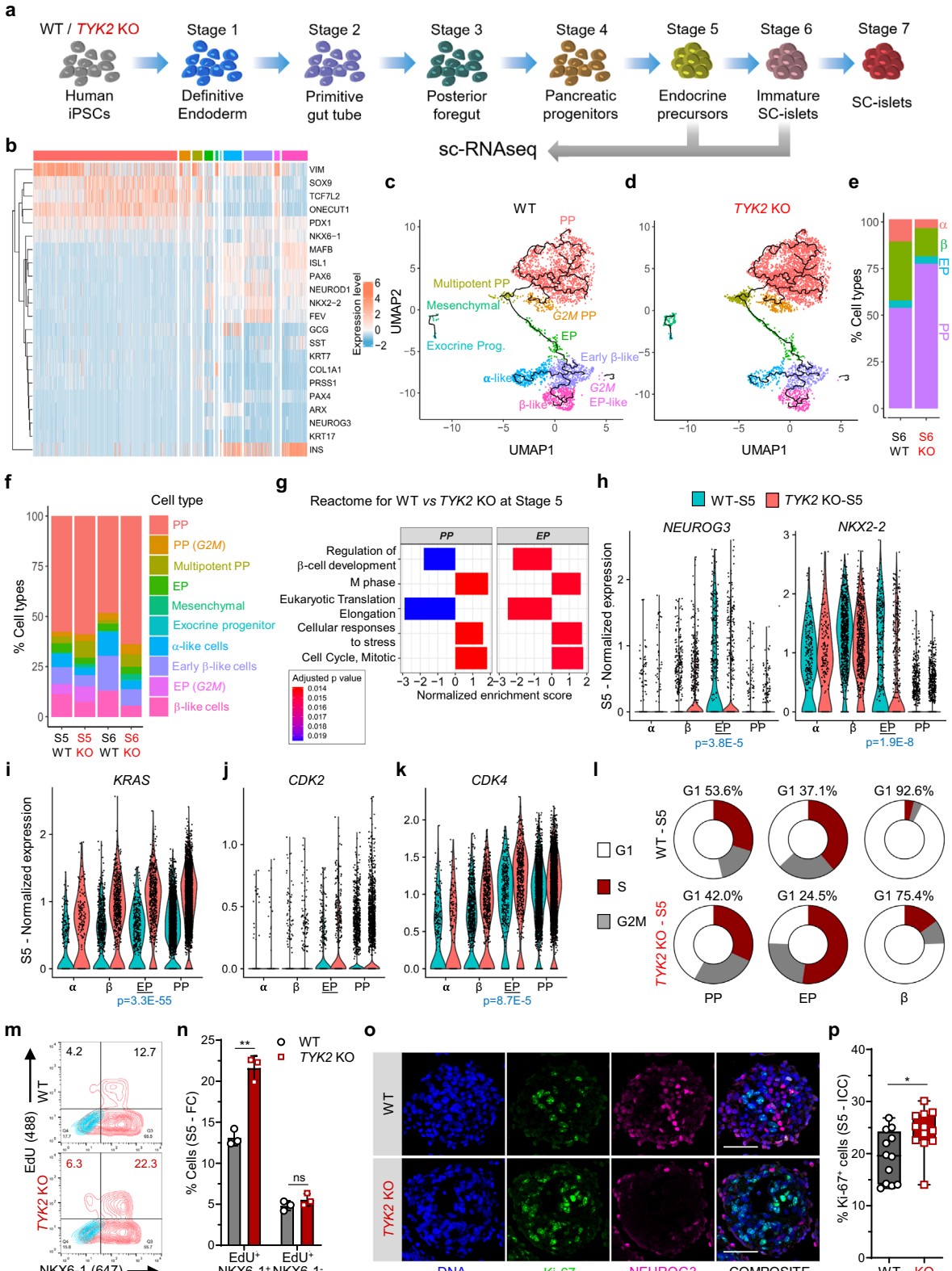

displayed limited IFNα pathway activation but preserved antiviral responses and β-cell function[10]. Our experiments showed that loss of TYK2 abolished the upregulation of ISGs including MHC Class I, equivalently in β-cells, α-cells, and their EPs (Fig. 5c). Additionally, the IFNα-induced upregulation of antigen processing genes, i.e., immunoproteasome (*PSMB8*, *PSMB9*) and peptide transporters (*TAP1*, *TAP2*, and *TAPBP*) were lost, suggesting impairment of the processing and

presentation of immunogenic peptides in the *TYK2* KO cells[27]. It is thus possible that these effects may synergize to inhibit autoimmune responses in vivo to a larger extent than what we observed in vitro using peptide-reactive CD8[+] T cells against peptide-pulsed SC-islets (Fig. 7). Despite the higher cytotoxic potency of these peptide-driven responses compared to the autoimmune responses against naturally processed and presented β-cell peptides, and the concomitant

**Fig. 4 | Single-cell transcriptomic analysis of endocrine differentiation.**
**a** Schematic for the scRNA-seq performed at Stage (S) 5 (3995 WT and 5025 KO -cells) and S6 (3348 WT and 4550 KO cells) samples. The figure was partly generated using Servier Medical Art, provided by Servier. **b** Heatmap for the selected genes associated with pancreatic differentiation. **c** Various noted clusters are indicated in different colour codes and presented as UMAP with pseudotime trajectories for S6 WT, **d** *TYK2* KO and **e** relative percentages abundance of pancreatic progenitors (PP), endocrine precursors (EP), β-like cells (β), and α-like cells (α). **f** Relative abundance of all noted clusters in S5 and S6 for WT and *TYK2* KO samples indicated in different colour codes. **g** Selected enriched pathways in S5 PP and EP like clusters using Gene set enrichment analysis (GSEA, Reactome). **h** Violin plots for the relative

expression of *NEUROG3* and *NKX2-2*; **i** *KRAS*, **j** *CDK2*, **k** *CDK4* in indicated α, β, EP, and PP like clusters. **l** Donut chart showing the percentage of cells at various cell cycle phases with Seurat (CellCycleScoring) pipeline in PP, EP, and β-like cells clusters. **m** Flow cytometry analysis for NKX6-1⁺EdU⁺ cells during S5 and **n** their quantification ($n = 3$). Data are means ± S.D. **o** Immunocytochemistry for NEUROG3 and Ki-67 expression at S5. Scale bar = 100 μm. **p** Quantification of the data in panel **o** presented from two experiments with six images each ($n = 12$). The box and whiskers plot showing the median with min to max whiskers. Two-tailed unpaired *t*-tests were performed to determine the significance levels for **n** and **p**. *$p < 0.05$; **$p < 0.01$; ns−non-significant. Source data are provided as a Source Data file.

inhibition of IFNα-induced PD-L1 upregulation, TYK2i was still effective at limiting SC-islet vulnerability to T-cells. Careful TYK2i titration, or concomitant targeting of other downstream mediators of IFNα signalling, may allow to dissociate its inhibitory effects on MHC Class I and PD-L1. Moreover, the IFNα induced upregulation of MHC Class II genes (*HLA-DQB1*, *HLA-DRB1*) was also rescued by TYK2 inhibition in this SC-islets model, suggesting that a downstream reduction in the activation of diabetogenic CD4⁺ T-cells may also be obtained.

A recent report suggests that iPSC-α cells are selectively protected from T-cell mediated destruction compared to iPSC-β cells following co-culture with autologous PBMCs, albeit they were immunogenic in an allogenic setting[40]. Importantly, our scRNA-seq dataset revealed a similar *TYK2* dependent upregulation of ISGs in α-cell- and β-cell-like populations, indicating a similar regulatory mechanism. The present model may thus prove useful to understand how α-cells are selectively protected from T-cell mediated autoimmunity despite having similar IFNα responses.

In summary, using *TYK2* KO SC-islet models, we deciphered the dual role of the candidate gene *TYK2* in pancreatic β-cells. First, depletion of TYK2 during early islet development affected the endocrine commitment, while it did not affect the functionality of mature beta cells. Second, TYK2 inhibition in mature islet cells effectively inhibited the IFNα induced upregulation of the antigen processing and presentation machinery, which reduced vulnerability to T-cell cytotoxicity. Inhibiting TYK2 signalling was sufficient to protect SC-islets from IFNα responses and T-cell cytotoxicity. Importantly, the TYK2i BMS-986165 used in this study is already into phase 2 and 3 trials for psoriasis[20,41] and may thus provide an attractive candidate for T1D interventions.

## Methods

### hPSCs cell lines and genome editing
hiPSCs line HEL46.11 (derived from human neonatal foreskin fibroblast)[42] and Human embryonic stem cell line H1 (WA01, WiCell) were used in the study. HEL46.11 hiPSCs were used to generate the *TYK2* knockout lines used in this study. The hiPSCs were generated and used according to the approval of the coordinating ethics committee of the Helsinki and Uusimaa Hospital District (no. 423/13/03/00/08). The hiPSCs were cultured on Matrigel (Corning, #354277) coated plates in Essential 8 (E8) medium (Thermo Fisher, A1517001) and passaged with 5 mM EDTA (Thermo Fisher, #15575-038) in PBS. The cell lines were routinely tested for the mycoplasma contamination.

For knocking out *TYK2* in HEL46.11 iPSCs, the ATG starting codon-containing third exon was deleted using two CRISPR/Cas9 guides that were designed with Benchling (Biology Software, 2017) (G1 AAGAGC-TAACAGGGGTCTCT and G2 GTCTGGGGCGTTGGCACCAT). Two million iPSCs were electoporated with 6 μg of CAG-Cas9-T2A-EGFP-ires-Puro (deposited in Addgene, plasmid no. 78311, together with detailed protocols for its use), 500 ng of gRNA1-PCR *TYK2* and 500 ng of gRNA2-PCR *TYK2* products, using Neon Transfection system (Thermo Fisher Scientific; 1100 V; 20 ms; two pulses). Single cells were sorted, expanded, and screened for ~300 bp deletion of exon 3. Positive clones were validated by Sanger sequencing. The top three CRISPR off-target

hits were sequenced and did not have any indels. Primers sequences used in the study to for genome editing is described in the Supplementary Table 2.

### In vitro hPSCs culture and their pancreatic lineage differentiation
hPSCs were differentiated towards pancreatic lineage to generate SC-islets using our recently published detailed protocol[15] with minor modifications. Briefly, near 80% confluent plates of stem cells were dissociated using EDTA and seeded on new Matrigel coated plates in E8 medium supplemented with 10 μM Rho-Associated kinase inhibitor (ROCKi, Y-27632, Selleckchem S1049) at a density of approximately 0.22 million cells/cm². The differentiation was started following 24 h of seeding and proceeded through a 7-stage differentiation protocol (stages 1–4 in adherent culture, stage 5 in AggreWell (Stemcell Technologies, #34421), and stages 6 and 7 in suspension culture).

The detailed differentiation protocol and the stage specific complete media formulations are described below. The details of the reagents used in the differentiation protocol described in the Supplementary Tables 3 and 4.

### Human pancreatic islets RNAseq for correlation studies
Data from human pancreatic islets were processed as described[19]. Briefly, human islets ($n = 191$) were obtained through the EXODIAB network from the Nordic Network for Clinical Islet Transplantation. 28 T2D donors had a clinical diagnosis of T2D. RNA was extracted using miRNeasy (Qiagen) or the AllPrep DNA/RNA (Qiagen) mini kits miR-Neasy (Qiagen) or the AllPrep DNA/RNA (Qiagen) mini kits. Library preparation of high-quality RNA (RIN > 8) was performed using the TruSeq RNA sample preparation kit (Illumina). Ethical approval of the collection of human pancreatic islets from organ donors has been obtained from the ethical committee in Lund (Dnr. LU 2011/263). All procedures in human islets were approved by the ethics committees at the Uppsala and Lund Universities and informed consent obtained by appropriate measures from donors or their relatives. The study design and conduct complied with all relevant regulations regarding the use of human study participants and was conducted in accordance with the criteria set by the Declaration of Helsinki.

### Human fetal pancreas
RNA was extracted from tissue biopsies of foetuses from terminated pregnancies (7–14 gestational weeks) using the TRIZOL method ($n = 16$). RNA libraries were then constructed using the TruSeq RNA library preparation kit (Illumina). RNA sequencing was performed on a HiSeq 2000 / Nextseq (Illumina). Paired-end 101 bp-long reads were aligned to the Reference Human Genome Build 37 using STAR and counts were generated as described[43]. Batch corrections were performed using the COMBAT function R package v3.32.1. Expression levels between fetal and adult (from GTEX) pancreas were compared in EdgeR v3.26.8. Gene expression was related to days post-conception using Pearson's correlation. Gene-gene correlations were performed using Pearson's correlation after log normalization of counts. Ethical permit for the collection of fetal pancreatic tissue has been obtained

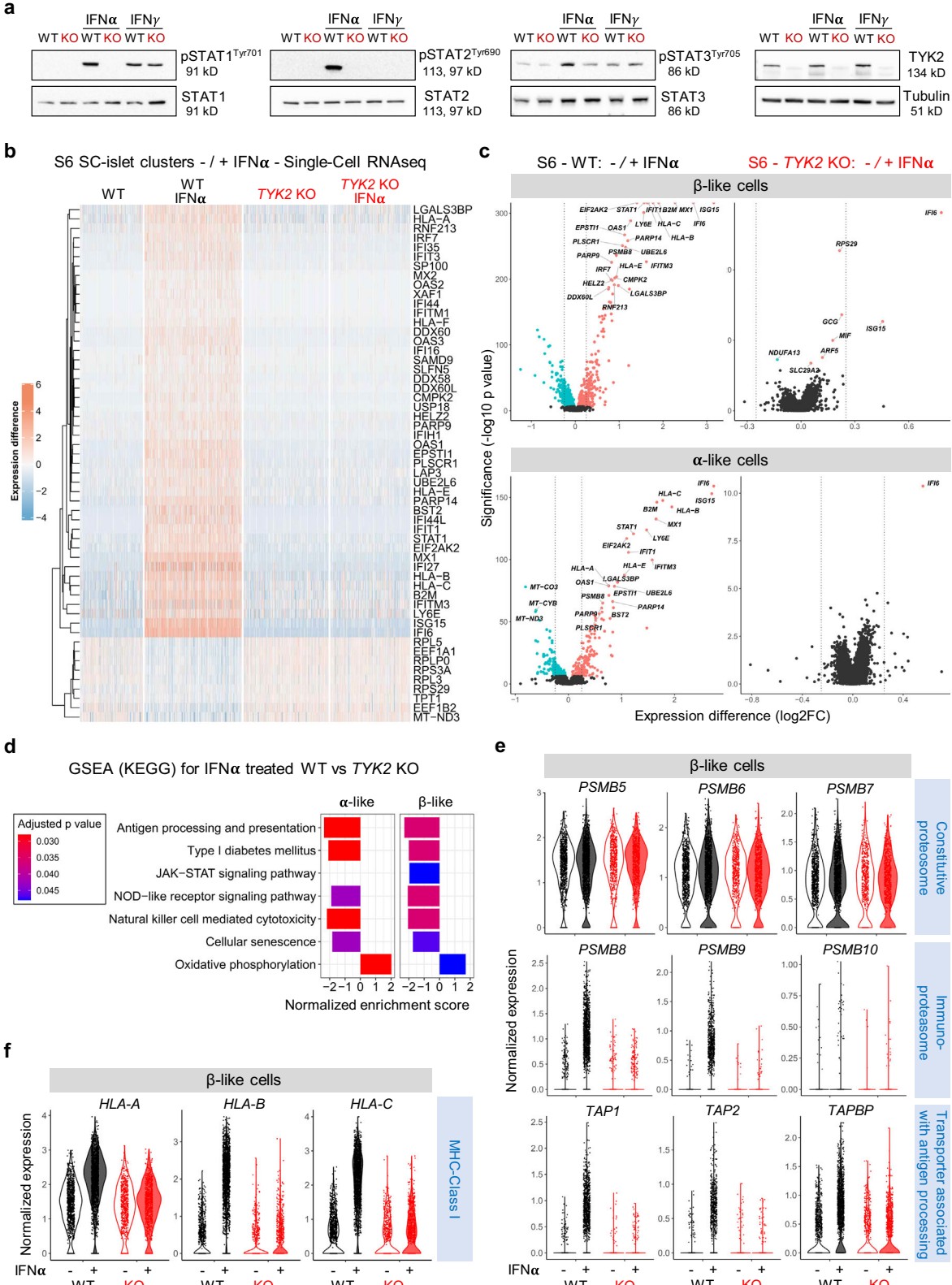

**Fig. 5 | TYK2 regulates the IFNα responses in SC-islets. a** Immunoblot analysis for the phosphorylation of STAT1, STAT2, and STAT3 in Stage (S) 6 SC-islets in response to either IFNα or IFNγ treatment (*n* = 3). **b** Heatmap showing the differentially expressed genes in the WT (3348 cells), WT + IFNα (5202 cells), KO (4550 cells), and KO + IFNα (4281 cells) samples. Single-cell transcriptomics performed following 24 h IFNα (100 ng/ml) treatment on the WT and *TYK2* KO S6 SC-islets. **c** Volcano plot showing the significant upregulated (soft Red), downregulated (iris

blue) and non-significant (black) genes in response to IFNα treatment in β-like cells and α-like cells. **d** GSEA (KEGG) for selected enriched gene sets shown in α- and β-like cells in response to IFNα treatment. **e** Violin plots showing the normalized expression of Constitutive proteasome genes, Immunoproteasome genes, Transporter associated with antigen processing genes and **f** MHC Class I genes in S6 β-like cells in response to IFNα. Source data are provided as a Source Data file.

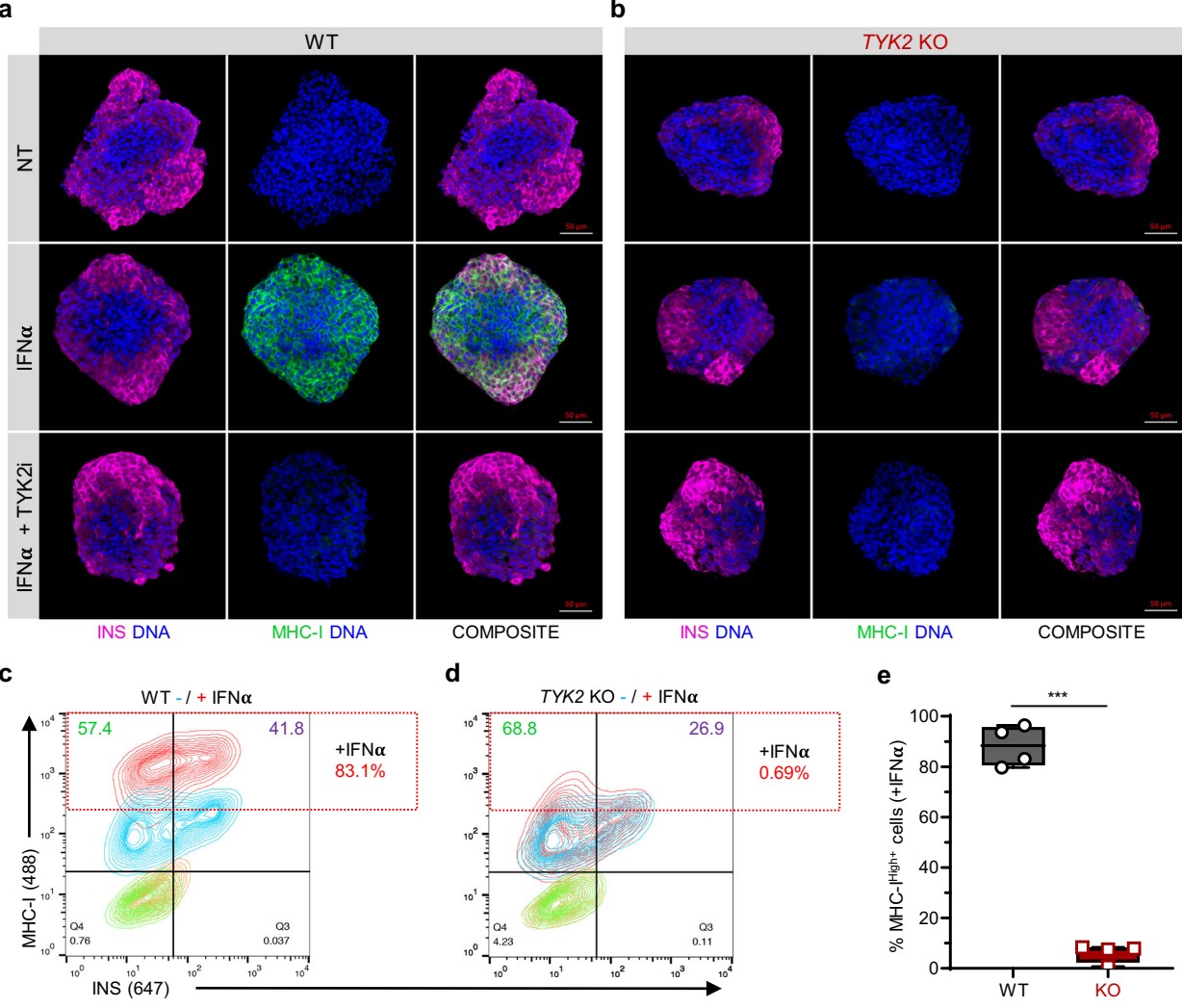

**a** WT

**b** *TYK2* KO

INS DNA    MHC-I DNA    COMPOSITE

INS DNA    MHC-I DNA    COMPOSITE

**c** WT − / + IFNα

**d** *TYK2* KO − / + IFNα

**e**

**Fig. 6 | TYK2 modulates MHC Class I presentation in SC-islets.**
**a**, **b** Representative images of Stage 7 SC-islets showing the immunocytochemistry for insulin (red) and MHC Class I (green) expression following 24 h treatment with IFNα alone or in combination with TYK2i (*n* = 3). Panel **a**, WT cells; panel **b**, *TYK2* KO cells. Scale bar = 50 μm. **c**, **d** Representative contour plots of flow cytometry showing the expression of insulin and MHC Class I following 24 h IFNα treatment. Panel **c**, WT cells; panel **d**, *TYK2* KO cells. **e** Quantification of the data in **c**, **d** (*n* = 4). Two-tailed unpaired *t*-test was performed to determine the significance levels. Box and whiskers plot showing median with min to max whiskers. \*\*\**p* < 0.001. Source data are provided as a Source Data file.

from the ethical committee in Lund (Dnr2012/593, Dnr 2015/241 and 2018/579). All procedures regarding human fetal pancreas were approved by the ethics committees at the Lund University and informed consent obtained by appropriate measures from relatives. The study design and conduct complied with all relevant regulations regarding the use of human study participants and was conducted in accordance with the criteria set by the Declaration of Helsinki.

**Ultra-deep bulk RNAseq analysis**
We performed ultra-deep RNAseq analysis for the HEL46.11 derived stage 1 (DE), stage 4 (PP) and stage 7 (SC-islets) samples shown in Fig. 1b, c. The differentiation, RNA isolation and library preparation was performed as described previously[12]. Briefly, total RNA from hiPSC-derived cells was isolated using NucleoSpin Plus RNA kit (Macherey-Nagel), treated with DNase I (NEB) and quantified using RNA HS Assay Kit (Qubit). Directional RNA-seq libraries were prepared using TruSeq Stranded mRNA Library Prep Kit (Illumina). Libraries were sequenced using Illumina HiSeq 2500 (with chemistry v4) at Eurofins Genomics.

The read pairs were mapped to the human reference genome (GRCh38) with STAR aligner[44]. Gene expression was counted from read pairs mapping to exons using featureCounts in Rsubread v2.4.3[45]. Duplicates, chimeric and multimapping reads were excluded, as well as reads with low mapping score (MAPQ < 10). The read count data was analysed with DESeq2 v1.30.1[46]. We analysed the effect of differentiation as a function of time, as well as pairwise comparisons between the different developmental stages (DE−PP−SC-islets). We removed genes with low expression levels from the analysis (<50 reads across all samples). PCA was calculated with prcomp using normalized counts that were scaled using the voom-function from limma[47].

For WT and *TYK2* KO genotype samples, RNAseq (Illumina HiSeq 2500; v4) was prepared as mentioned above and performed at stage 0 (iPSCs), stage 4 (PP), stage 5 (EP), and stage 6 (EC) as presented in Fig. 3. The raw data was filtered with Cutadapt v2.6 to remove adapter sequences, ambiguous (N) and low-quality bases (Phred score <25). We also excluded read pairs that were too short (<25 bp) after trimming. The filtered read pair were mapped to the human reference genome (GRCh38) with STAR aligner v2.5.4b[44] and processed as described above. We analysed the effect of *TYK2* knockout separately for each developmental stage (iPSCs, stage 4, stage 5, and stage 6). For false discovery rate (FDR) estimation we used Fdrtool v1.2.17[48]. The differentially expressed genes (FDR<0.01) were analysed for enrichment

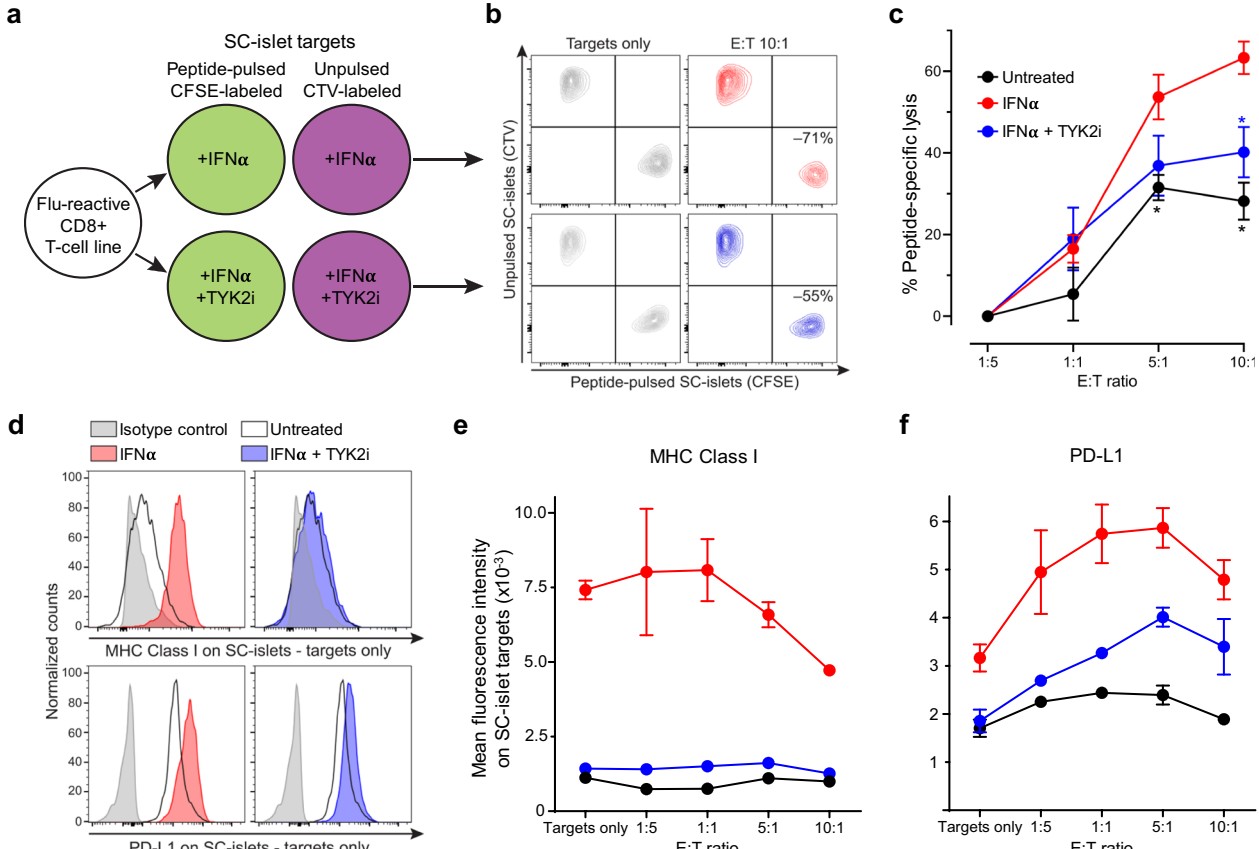

**Fig. 7 | TYK2 inhibition modulates T-cell-mediated cytotoxicity in SC-islets.**
**a** Schematic of T-cell/SC-islet co-culture experiments. Increasing numbers of flu peptide-reactive CD8⁺ T cells were incubated for 6 h with a fixed mixture of peptide-pulsed, CFSE-labelled SC-islets and unpulsed, CellTrace Violet (CTV)-labelled SC-islets, which were preliminary treated for 24 h with IFNα alone or in combination with TYK2i or left untreated. The ratio of surviving (Live/Dead-negative) pulsed CFSE⁺ vs. unpulsed CTV⁺ SC-islets thus provided a readout of peptide-specific lysis. **b** Representative flow cytometry contour plots of IFNα-treated (top) and IFNα/TYK2i-treated SC-islets (bottom), cultured alone (left) or with T cells at an effector-to-target (E:T) ratio of 10:1 (right). Each population is gated on Live/Dead-negative

events. The percent peptide-specific lysis in the presence of T cells (i.e., the ratio of CFSE⁺/CTV⁺ live cells normalized to the ratio in the absence of T cells) is indicated. **c** Peptide-specific lysis for the indicated conditions at different E:T ratios. Data are normalized means ± SEM of two experiments performed in triplicates: *$p < 0.05$ by two-tailed Mann–Whitney $U$ test vs IFNα. **d** Representative flow cytometry histograms of MHC Class I (top) and PD-L1 expression (bottom) in SC-islet targets treated with IFNα (left) or IFNα/TYK2i (right). **e**, **f** Mean fluorescence intensity of **e** MHC Class I and **f** PD-L1 expression in SC-islets at different E:T ratios. Data are means ± SEM of duplicate wells and one representative experiment out of two performed is shown. Source data are provided as a Source Data file.

separately for the up- and downregulated genes using ClusterProfiler v3.18.1[49] against the Reactome pathways[50].

## Single cell RNA sequencing sample preparation and analysis

WT and *TYK2* KO genotype stage 5 and stage 6 (either untreated or with 100 ng/ml for 24 h IFNα treatment) samples were prepared for the single cell RNAseq analysis as we described previously[15]. Briefly, cells were dissociated with 2 ml of a 1:1 mixture of TrypLE Select (Thermo catalog no. 12563-029) and Trypsin-EDTA (Sigma catalog no. T4174, 10× stock diluted 1:10 with PBS) for 10 min at 37 °C. Dissociation agent was neutralized by adding 12 ml of ice-cold 5% FBS-PBS. Cells were passed through a 30 μm strainer (BD) to remove cell clumps, centrifuged at 200 rcf for 5 min, washed twice in encapsulation buffer (1× PBS with 0.04% BSA), counted and adjusted to a 1 × 10⁶/ml cell suspension for encapsulation.

Single cell gene expression profiles were generated with 10x Genomics Chromium Single Cell 3′RNAseq platform using the Chromium Next GEM Single Cell 3′ Gene Expression (version 3.1 chemistry). Raw data (fastq) processing was performed with 10x Genomics Cell Ranger (v3.1) pipeline. The reads were mapped to the human reference genome (GRCh38.98). The filtered counts were analysed with Seurat v3.2.3[22]. The counts were normalized, scaled, and analysed for PCA with default methods. The variable genes (top 1000) were identified

separately for each sample and combined during the analysis (for a total of 1199 variable genes). To reduce biases among datasets we used Harmony v1.0[51] on the first 50 PCs with sample as the covariable (with theta = 2, nclust = 50, max.iter.cluster = 40, max.iter.harmony = 10). The integrated (harmonized) PCs were used to build the Uniform Manifold Approximation and Projection (UMAP), find the neighboring cells (using Shared Nearest Neighbor), and identify cell clusters using default Seurat methods. To reduce background RNA contamination from disrupted cells we used SoupX v1.4.8[52] with clusters identified with Seurat, and known cell type specific marker genes (*GCG*, *TTR*, *INS*, *IAPP*, *SST*, *GHRL*, *PPY*, *COL3A1*, *CPA1*, *CLPS*, *REG1A*, *CELA3A*, *CTRB1*, *CTRB2*, *PRSS2*, *CPA2*, *KRT19*, and *VTCN1*) to estimate the level of contamination. The Seurat analysis was then repeated with the adjusted counts with the following modifications. Cells with less than 5000 UMI counts or 1700 expressed genes were excluded. We also removed cells with unusually high level of mitochondrial reads (>20% of counts). We assigned a cell cycle phase to all the cells using the default settings in CellCycleScoring-function and used this to remove biases due to cell cycle heterogeneity when scaling the data. During clustering the resolution was set to 0.2. Differentially expressed genes among clusters and sample types were identified with Wilcox-test using Seurat. The clusters were reordered by similarity and identified to cell types by the differentially expressed genes corresponding to known marker

genes. The differentially expressed genes (FDR < 0.01) were analysed for enrichment separately for the up- and downregulated genes using ClusterProfiler v3.18.1[49] against the KEGG. ClusterProfiler was also used for geneset enrichment analysis (GSEA) against the same database, using fold change to rank the genes.

### In vivo animal transplantation studies

Animal care and experiments were approved by National Animal Experiment Board in Finland (ESAVI/14852/2018). NOD-SCID-Gamma (NSG, 005557, Jackson Laboratory) mice were obtained from SCAN-BUR and housed at Biomedicum Helsinki animal facility, on a 12 h light/dark cycle and fed standard chow ad libitum, 2016 Teklad global 16% protein rodent diets (ENVIGO). The temperature was kept at 23 °C with 24 relative humidity (RH). Implantations were performed on 3- to 8-month-old mice as described previously[15]. Briefly, stage 7- WT and *TYK2* KO SC-islets equivalent to approximately 2 million cells were loaded in PE-50 tubing and implanted under the kidney capsule. Mouse serum samples were collected monthly from the saphenous vein and stored at −80 °C for human C-peptide analysis. Human-specific C-peptide was measured from plasma samples with the Ultrasensitive C-peptide ELISA kit (Mercodia, Uppsala, Sweden).

### Overexpression of TYK2 WT protein in endocrine precursors cells

WT-TYK2 was transiently overexpressed during S5 in WT and KO differentiated cells. The pRC/CMV-TYK2-VSV plasmid was obtained from Dr. Sandra Pellegrini (Addgene #139344)[53]. At the end of S4, cells were dissociated into single cell suspension using TrypLE (Thermo Fisher Scientific) for 10 min at 37 °C and $5 \times 10^6$ ells were electroporated with 2 μg of the plasmid using Neon Transfection system (Thermo Fisher Scientific; 1100 V; 20 ms; two pulses). Cells were then plated in AggreWell plates to form 3D aggregates for 24–48 h.

### Western blot

For protein extraction, cells were washed with ice-cold PBS and lysed with Cell lysis buffer (Cell Signaling Technologies #9803) for 10 min on ice. The cells were sonicated for 3 × 5 s on ice, centrifuged (1000 rcf for 10 min at 4 °C) and the supernatant was stored at −80 °C. The samples were run on Any kD Mini-PROTEAN TGX gel (Bio-Rad Laboratories) and then dry transferred onto a nitrocellulose membrane using the iBlot system (Invitrogen) as per manufacturer's instructions. The membrane was then probed with the primary antibody overnight at 4 °C, washed twice with Tris-buffered saline containing 0.05% Tween for 2 × 10 min, and incubated with the corresponding secondary antibody for 30 min at room temperature. Chemiluminescence detection was performed with Amersham ECL (RPN2235; Cytiva) and Bio-RAD Chemidoc XRS1 imaging system; Image Lab software v6.0.0. The details of antibodies and their dilutions used in the study for WB are described in the Supplementary Table 5.

### Flow cytometry

For quantifying the definitive endoderm positive cells in stage 1, cytometry for CXCR4 was performed as previously described[12]. Briefly, the cells were incubated in room temperature for 30 min with 1:10 PE Mouse anti-human CXCR4 antibody (BD Biosciences) or the PE Mouse IgG2a, kappa isotype before analysis with FACSCalibur (BD Biosciences). For intracellular antigen cytometry, cells were dissociated with TrypLE (Thermo Fisher Scientific) for 6 min at 37 °C and resuspended in cold 5% FBS-containing PBS. Cells were fixed and permeabilized using Cytofix/Cytoperm (554714, BD Biosciences) as per manufacturer's instructions. Primary or conjugated antibodies were incubated with the cells overnight at 4 °C in Perm/Wash buffer (554714, BD Biosciences) containing 4% FBS and secondary antibodies for 45 min at RT. The cells were then analysed using FACSCalibur cytometer (BD Biosciences) with BD Cellquest Pro v4.0.2 (BD Biosciences)

and FlowJo (Tree Star Inc.) softwares. For EdU and NKX6-1 double staining, Click-iT EdU Flow cytometry assay kit (Thermo Fisher Scientific, #C10419) was used. S5 cells were labelled with 5 μM EdU for 18 h and then cells were harvested and stained for EdU and intracellular NKX6-1 as per manufacturer's instructions. For all the flow cytometry samples, cells were gated with FSC and SSC to remove cellular debris. Positive and negative gating was determined through negatively stained cells within the same population and/or non-stained conjugated IgG isotype controls. For the T-cell cytotoxic assay, after excluding CFSEnegCTVneg CD8+ T cells, SC-islet cells were gated on viable singlets and then on CFSE+ and CTV+ fractions to calculate their relative proportion. The details of antibodies and their dilutions used in the study for flow cytometry are described in the Supplementary Table 6.

### mRNA extraction and qRT-PCR

Total RNA from hiPSC-derived cells was isolated using NucleoSpin Plus RNA kit (Macherey-Nagel). A total of 1.5 μg RNA was reversely transcribed using Moloney murine leukemia virus reverse transcriptase (M1701, Promega) for 90 min at 37 °C. In all, 50 ng cDNA was amplified using 5x HOT FIREPol EvaGreen qPCR Mix Plus no ROX (Solisbiodyne) in a 20 μl reaction. The reactions were pipetted using QIAgility (Qiagen) robot into 100 well disc run in Rotor-Gene Q. Relative quantification of gene expression was analysed using ΔΔCt method, with cyclophilin G (PPIG) as a reference gene. Reverse transcription without template was used as negative control and exogenous positive control was used as a calibrator. The qRT-PCR primers sequence will be made available from lead contact upon request.

### Immunocytochemistry and immunohistochemistry

For paraffin embedding, aggregates were fixed with 4% PFA at 4 °C for 24 h, following eosin staining, aggregates were embedded in 2% low-melting agarose (Fisher Bioreagents) PBS and transferred to paraffin blocks. WT or *TYK2* KO implanted grafts were retrieved at 2 months, dissected, and fixed with 4% PFA at RT for 48 h, paraffin embedded, and cut into 5 μm sections using Leica microtome. For immunohistochemistry, slides were deparaffinized and antigens retrieved by boiling slides in 0.1 M citrate buffer (pH 6) using Decloaking chamber (Biocare Medical) at 95 °C for 12 min. For whole mount or adherent cultures staining, cells were fixed in 4% PFA for 15 min at RT, permeabilized with 0.5% Triton X-100 in PBS for 15 min at RT, then blocked with Ultra-V (Thermo Fisher Scientific) for 10 min and incubated with primary antibodies overnight at 4 °C and secondary antibodies for 1 h at RT diluted in 0.1% Tween in PBS. Invitrogen™ EVOS™ FL digital inverted fluorescence microscope or Zeiss Axio Observer Z1 with Apotome were used to image the cells and further processed with ZEN-2 software. All stained samples were equally treated and imaged with the same microscope parameters. Image quantification was performed using CellProfiler software[54] and Fiji software[55]. The details of antibodies and their dilutions used in the study for immuno-cytochemistry are described in the Supplementary Table 6.

### Static and dynamic glucose-stimulated insulin secretion

For the static assay, fifty aggregates were manually picked and preincubated in 2.8 mmol/L glucose Krebs buffer in a 12-well plate placed on a 95-rpm rotating platform for 90 min at 37 °C. Aggregates were then washed with Krebs buffer and sequentially incubated in Krebs buffer containing 2.8 mmol/L glucose, 16.6 mmol/L glucose, and 2.8 mmol/L glucose plus 30 mmol/L KCL, for periods of 30 min each. Samples of 200 μL were collected from each treatment and stored at −80 °C for insulin ELISA measurements. Dynamic tests of insulin secretion were carried out using a perifusion apparatus (Brandel Suprafusion SF-06, MD, USA) with a flow rate of 0.25 ml/min and sampling every 4 min. Samples from each fraction collected were analysed using insulin ELISA (Mercodia, Sweden). Following static and

dynamic tests of insulin secretion, the SC-islets were collected, and the total insulin and DNA contents were analysed. Stimulated insulin secretion results are normalized using total DNA content.

## Cytotoxicity assays

A CD8+ T-cell line reactive to the HLA-A2-restricted peptide Influenza virus matrix protein MP$_{58-66}$ (32% peptide-specific by tetramer staining) was thawed and rested for 3 h before use. Meanwhile, islet clusters were dissociated with TrypLE, stained with 1 µM CFSE or CellTrace Violet (CTV) and incubated for 2 h with 0.1 µM Influenza MP$_{58-66}$ or peptide diluent, respectively. After washing, CFSE- and CTV-labelled SC-islets were mixed in equal numbers and cultured for 6 h in triplicate at $1 \times 10^5$ cells/each per well in 96-well flat-bottom plates, alone or with increasing numbers of T cells ($0.2 \times 10^5$, $1 \times 10^5$, $5 \times 10^5$, and $10 \times 10^5$), corresponding to effector-to-target (E:T) ratios of 1:5, 1:1, 5:1, and 10:1, respectively. After washing, cells were stained with Live/Dead Fixable Far Red (Thermo Fisher), antibodies against HLA-A, B, C (RRID: AB_2566151) and PD-L1 (RRID: AB_940368) and acquired on a BD LSRFortessa flow cytometer. SC-islets were analysed by FlowJo after gating on Live/Dead⁻ events and separation of CFSE+ (peptide-pulsed) and CTV+ (unpulsed) populations. Percent peptide-specific lysis at different E:T ratios is expressed as the ratio of live CFSE+/CTV+ cells normalized to the same ratio in wells containing SC-islet targets alone.

## Quantification and statistical analysis

Data are collected from at least three independent differentiation experiments of 2 independent *TYK2* KO hiPSCs clones (C10 and C12). Blinding was applied for immunohistochemical quantification. Morphological data represents population-wide observation from independent differentiation experiments. All the representative images in the figures were reproducible in all the experiments performed and are followed by a quantification panel. Box and whiskers plots are presented as min to max showing all the points. Statistical methods used are described in each figure legend and individual method section. Briefly, Student's unpaired two-tailed *t* test with Welch correction was used to compare differences between two groups while for more than two groups one-way ANOVA followed by Tukey's test applied using Prism 8 software (GraphPad Software, La Jolla, CA). The results are presented as the mean ± S.D unless otherwise stated. The box-and-whisker plots are showing the median with lower and upper hinges corresponding to the first and third quartiles (the 25th and 75th percentiles) with min to max whiskers for range between minimum to maximum values. *P*-value < 0.05 were considered statistically significant (*$P < 0.05$; **$P < 0.01$; ***$P < 0.001$).

## Reporting summary

Further information on research design is available in the Nature Research Reporting Summary linked to this article.

## Data availability

Ultra-deep bulk RNAseq data for pancreatic differentiation stages −1, −4, and −7 of HEL46.11 and for stages −0, −4, −5, and 6 of WT and *TYK2* KO genotypes are deposited in the Gene Expression Omnibus database with accession code GSE190727 and GSE190725. Single-cell RNA-seq data for WT and *TYK2* KO genotype pancreatic differentiation stages −5, −6 and stage 6 with treatment of ±IFNα) are deposited in the Gene Expression Omnibus database with accession code GSE190726. Source data are provided with this paper. Original western blot images are deposited at Mendeley (https://data.mendeley.com/datasets/8n9nytgy57/1) and the source data file. Microscopy image acquisition data are reported in the source data file. Any additional information required to reanalyse the data reported in this paper is available from the source data files and the lead contacts on request, Vikash Chandra (vikash.chandra@helsinki.fi) and Timo Otonkoski (timo.otonkoski@helsinki.fi). Source data are provided with this paper.

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

## Acknowledgements

We gratefully acknowledge Dr. Fatoumata Samassa for help with cytotoxicity assays. We thank Jarkko Ustinov for the insulin and c-peptide ELISA measurements. S. Eurola, H. Grym, and A. Laitinen are thanked for expert technical support. We thank FIMM Single Cell Analytics unit (supported by HiLIFE and Biocentre Finland) for single cell RNA sequencing services. We want to acknowledge the participants and investigators of the FinnGen study. T.O. acknowledges the funding provided by the Academy of Finland (MetaStem Center of Excellence grant 312437), the Novo Nordisk Foundation and the Sigrid Juselius Foundation. R.M. acknowledges the support of the *Agence Nationale de la Recherche* (ANR-19-CE15-0014-01) and the *Fondation pour la Recherche Medicale* (EQU20193007831). C.H. was funded by an *Année Recherche* fellowship of the Paris Saclay University. F.V. was funded by an international PhD fellowship of the IdEx Université de Paris. R.B.P. acknowledges the funding by Hjelt foundation, Crafoord foundation (2020089) and Swedish Research Council (2021-02623). D.L.E. acknowledges the support of grants from the Welbio-FNRS (Fonds National de la Recherche Scientifique; WELBIO-CR-2019C-04), Belgium; the Innovate2CureType1—Dutch Diabetes Research Foundation (DDRF), Holland; the Juvenile Diabetes Foundation (JDRF; 2-SRA-2019-834-S-B); the NIH (HIRN-CBDS) grant U01 DK127786, USA. D.L.E., T.O., and R.M. acknowledge support from the Innovative Medicines Initiative 2 Joint Undertaking under grant agreements No 115797 (INNODIA) and 945268 (INNODIA HARVEST), supported by the European Union's Horizon 2020 research and innovation programme. These Joint Undertakings receive support from the Union's Horizon 2020 research and innovation programme and "EFPIA", "JDRF" and "The Leona M. and Harry B. Helmsley Charitable Trust".

## Author contributions

V.C. conceived and conceptualized the study, performed experiments, analysed data, and wrote the first draft. H.I. carried out the experiments, standardized the differentiation, analysed data, and participated in manuscript writing. C.H. and F.V. carried out the T-cell cytotoxicity assays and analysed the data. R.B.P., O.P.D., and L.G. performed Gene expression correlation analysis with fetal pancreas and adult islets

RNAseq datasets and analysed the data. C.H., F.V., and R.B.P. have contributed equally as second author, J.K. performed and analysed all the bulk and single cell RNAseq datasets. D.B. helped in generation of *TYK2* KO hiPSCs lines. J.S.V. and H.M. performed and analysed animal and differentiation experiments. T.B. and V.L. participated in the differentiation experiments and their analysis. I.A. provided and analysed human fetal RNAseq derived data. S.G. participated in microscopy and manuscript writing. R.M. conceptualized and supervised the T-cell cytotoxicity assays, analysed data, participated in manuscript writing, and acquired funding. D.L.E. conceived and supervised the study, acquired funding, and participated in manuscript writing. T.O. conceived and supervised the study, provided resources, acquired funding, and wrote the manuscript.

## Competing interests

D.L.E. received grant support from Eli Lilly and Company, Indianapolis, for research on new approaches to protect pancreatic beta cells in T1D. The remaining authors declare no competing interests.
