## [Peer Review File · Nature Communications]

The type 1 diabetes gene TYK2 regulates β -cell development and its responses to interferon- αREVIEWER COMMENTS

Reviewer #1 (Remarks to the Author):

In the manuscript titled “The type 1 diabetes gene TYK2 regulates beta cell development and its responses to interferon alpha” Chandra and Ibrahim et al. examine the role played by Tyk2 in beta cell development and in T1D. For this purpose, they KO the gene in an hESC cell line, and confirm their findings with chemical inhibition.

Their major findings are:

- 1) TYK2 KO hESCS are capable of differentiating into beta cells with similar quality to that of WT cells, albeit in smaller numbers due to reduced endocrine differentiation.
- 2) TYK2 mediates beta cells' response to ifna, and therefore in its absence beta cells are less prone to CD8 mediated killing.

Overall, the author's claims are well supported by the presented results. Tyk2's role in endocrine differentiation is novel, but its role as a modulator of the immune response is well known. And yet methodologically, the authors use of hESC-derived beta cells to study the potential mechanisms through which specific genes contribute to the eruption of T1D is original, and will likely inspire similar studies in the future.

Major issues:

- 1) The scRNA-seq data is presented in an obscure fashion, which limits the ability to criticize it. In this framework:
 - a. Violin plots, rather than boxplots, should be used to present gene expression across cell types, so that the reader can get an impression about the relative proportion of cells which express (or do not express) a gene.
 - b. One should be able to visually compare the WT and KO samples in details, rather than the general UMAP in figure 4c-d. the UMAP in figure 4d should be broken into one which includes on WT or only KO cells, so that a clear impression of the differences between populations is received. In addition, it is necessary to add a plot which compares the number of cells from each sample, for each cell type (e.g. violin plots). It seems like the KO sample contains two populations (acinar and proliferating Eps) that do not exist in the WT, and this may lead to different interpretations of the results altogether.
 - c. The authors do not mention the effect of TYK2 KO on the differentiation of non-pancreatic endocrine cells (enterochromaffin cells) which typically appear in vitro. In addition, they should elaborate about the effect of the KO on alpha cells.

Additional issues:

- 1) In figure 3C the authors choose to focus on KRAS, yet there are other genes with a significantly higher difference. The authors should at least mark their names.
- 2) In figure 4e the relevant text exclaims: “WT S6 contained 62.2% β -like cells compared to 37.7% in KO, whereas S6 KO 187 had still 64.4% PP-like cells compared to 35.5% in WT”. This description is misleading, as it suggests that in KO cells 37.7% are beta, whereas 64.4% are PP-like which is clearly more than 100%. Rather, it should be said: Among the beta cells analyzed, 62.2% were WT and 37.7% KO. Given this, the authors should also present a breakdown of each sample into the cell types which comprise it. This will likely uncover additional cell types which the authors did not mention, and if so – they should present them in a clear fashion.
- 3) In figure 4K the authors claim that G1 is shorter in TYK2 KO. However, since they did not show that the overall duration of the cell cycle is unchanged, it might just as well be that S is now longer.

Reviewer #2 (Remarks to the Author):

The type 1 diabetes gene TYK2 regulates beta cell development and its responses to IFN α

Chandra, Brahim et al.,

The authors studied the role of TYK2 in human induced pluripotent stem cells during beta cell development and response to IFN α using TYK2 ko iPSCs.

The manuscript is well written and the proper controls were used throughout the experiments presented. A few comments:

- How come that TYK2 ko had less ins mRNA, less ins content but still respond to glucose stimulation with same levels of ins as WT?
- Could it be that TYK2 ko is affecting other endocrine cells?
- What is the rationale of stimulating progenitor cells with IFN α ?
- Have you looked HLA-II genes? It would be interesting to see if TYK2 ko also prevents HLA-II upregulation, since it has been recently reported that HLA-II can be expressed in beta cells and could have a role in T1D pathogenesis DOI: 10.1007/s00125-021-05619-9
- It has been reported that Neurog3+ was found at higher levels in epsilon cells, how do you explain this in relation to your recent findings? <https://doi.org/10.3389/fendo.2021.736286>
- Would TYK2 inhibition prevent also an attack from the autoreactive CD8 T cells already present in the tissue?

Reviewer #3 (Remarks to the Author):

Chandra et al. studied the role of T1D associated gene TYK2 in β -cell development using the hPSC-based model. They found that TYK2 deletion led to the decreasing of endocrine precursors by regulating KRAS expression. Additionally, they showed that TYK2 knockout or inhibition prevented IFN α -induced antigen presentation and enhanced cell survival against CD8+ T-cell cytotoxicity. In general, this work is interesting, and provide a promising candidate for T1D treatment. However, several key questions need to be addressed:

Major concerns:

1. Based on the conclusions of this work, TYK2i BMS-986165 is a promising T1D drug candidate. It is necessary to test the in vivo function of this small molecule in diabetic mouse model, which will provide broader interests for the field.
2. Rescue experiments are required for TYK2 KO.
3. How about the phenotypes of hiPSCs with SNPs (rs34536443 and rs2304256)? Can the authors directly generate hiPSCs from patients with SNPs or generate hiPSCs with these SNPs using CRISPR?
4. How about the phenotypes of overexpression of TYK2?
5. The complete loss of TYK2 impaired EP formation thus reduced the number of INS+NKX6-1+ cells (~ 30% reduction), but why the ratio of INS+ cells were comparable in WT and TYK2 KO 2-month grafted tissue sections.
6. KRAS can influence the G1/S transition, only use the expression level of cell cycle markers to estimate the fraction of dividing cells in WT and KO groups is not sufficient. The

fraction of G1 and S phase cells should be determined by experiments, such as FACS.

Minor concerns:

1. Fig. 2h showed the total insulin content decreased in KO cells. It is better to show the specific insulin content value.
2. Fig. 3b, although the p-value is significant, the difference between WT and KO samples at S5 is too modest.
3. The authors showed TYK2 KO could significantly decrease the INS+ cells at S6 in Figure 2 and Figure 4, but the INS+ cells showed no difference between WT and KO cells in Fig. 6a and b. This should be consistent.
4. In Fig. 4i, j, whether the expression level of CDK2 and CDK4 is significant in PP and EP cells? There should be marked in the figure.
5. All bioinformatics tools used in this paper have no version information, such as featureCounts, Cutadapt. Detailed information should be provided.
6. In line 402 to 404, 'Raw sequencing data...using the Pearson method'. This sentence seems redundant.
7. The criteria for filtering low expression genes using reads count seems unreliable for short or long genes that probabilistically have less or more reads coverage because of gene length. Therefore, it would be recommendable to use normalized gene expression (e.g., FPKM, TPM) for filtering.
8. In line 435, for definition of differentially expressed genes, the author didn't note the threshold for expression fold change. And the DEGs showed in Fig. 3c seems have even lower p-values but not 0.01.
9. In line 156-158, the author indicated that NEUROG3 and NKX2-2 were significantly down-regulated at S5. However, the author didn't show the significance test (e.g., Expression fold change and FDR calculated by DESeq2).
10. In Figure 3f, the author showed the results of Reactome enrichment analysis but titled with GO enrichment. Additionally, x axis should note with percentage.

Reviewer #1 (Remarks to the Author):

In the manuscript titled “The type 1 diabetes gene *TYK2* regulates beta cell development and its responses to interferon alpha” Chandra and Ibrahim et al. examine the role played by Tyk2 in beta cell development and in T1D. For this purpose, they KO the gene in an hESC cell line, and confirm their findings with chemical inhibition.

Their major findings are:

- 1) TYK2 KO hESCS are capable of differentiating into beta cells with similar quality to that of WT cells, albeit in smaller numbers due to reduced endocrine differentiation.
- 2) TYK2 mediates beta cells' response to ifna, and therefore in its absence beta cells are less prone to CD8 mediated killing.

Overall, the author's claims are well supported by the presented results. Tyk2's role in endocrine differentiation is novel, but its role as a modulator of the immune response is well known. And yet methodologically, the authors use of hESC-derived beta cells to study the potential mechanisms through which specific genes contribute to the eruption of T1D is original and will likely inspire similar studies in the future.

We would like to thank Reviewer 1 for his/her clear and focused feedback on our manuscript and appreciation that our model that could inspire similar studies for other T1D candidate genes. We hope the revision outlined below will further strengthen our manuscript and clarify our novel findings.

Major issues:

1) The scRNA-seq data is presented in an obscure fashion, which limits the ability to criticize it. In this framework:

a. Violin plots, rather than boxplots, should be used to present gene expression across cell types, so that the reader can get an impression about the relative proportion of cells which express (or do not express) a gene.

We have now changed all the boxplots to violin plots in Fig. 4, Fig. 5 and Supplementary Fig. 8.

b. One should be able to visually compare the WT and KO samples in details, rather than the general UMAP in figure 4c-d. the UMAP in figure 4d should be broken into one which includes on WT or only KO cells, so that a clear impression of the differences between populations is received.

The point is well taken. We have now broken the previous Fig. 4d into new Fig. 4c (WT) and 4d (*TYK2* KO) stage 6 (S6) UMAP profile and updated the text correspondingly in the Result section. Moreover, we have now added a bar plot for the percentage of major cell types at S6 (Fig. 4e) and for all cell types in S5 and S6 in WT and KO samples as Fig. 4f.

In addition, it is necessary to add a plot which compares the number of cells from each sample, for each cell type (e.g., violin plots).

We have now added a plot (Supplementary Fig. 6d, e) to compare the exact and relative (percentage) number of cells from each sample in each cell type as suggested.

It seems like the KO sample contains two populations (acinar and proliferating Eps) that do not exist in the WT, and this may lead to different interpretations of the results altogether.

Thank you for pointing this out. Here, we would like to clarify that, in the superimposed color-coded image of all samples (old Fig. 4d), it appears that these two populations do not exist in the WT. But, as suggested, we now have presented a separate UMAP for WT and *TYK2* KO samples of S5 and S6 (New: Fig. 4c, d; Supplementary Fig. 6b, c) which indicates the existence of both populations in WT and KO

albeit at different percentage levels (Supplementary Fig. 6d, e). The exocrine progenitors and multipotent PPs have now been correctly marked on the new UMAPs.

c. The authors do not mention the effect of *TYK2* KO on the differentiation of non-pancreatic endocrine cells (enterochromaffin cells) which typically appear *in vitro*. In addition, they should elaborate about the effect of the KO on alpha cells.

This is a very good point. Indeed, we and others^{1,2} have described stem cell-derived enterochromaffin-like cells (SC-EC) with SC-islet differentiation. We have now performed additional immunostaining for the SC-EC marker, SLC18A1 and INS on mature S7 WT and *TYK2* KO SC-islets and quantified them (Fig. 2j, k) (lines: 119-122). We do observe the SC-EC cells in our differentiation and interestingly we observed their significant ($\approx 31\%$) reduction in *TYK2* KO SC-islets. Notably, the SC-ECs follow similar *in vitro* development trajectory like β -cells².

Additionally, to understand the effect of *TYK2* KO on other endocrine cell types e.g., the α - and δ - cells, we performed the immunostaining of GCG and SST on mature S7 WT and *TYK2* KO SC-islets. We did not observe any significant differences in either α - or δ - like cell population between WT and *TYK2* KO S7 SC-islets (Supplementary Fig. 3a, b) (lines: 117-120).

Additional issues:

1) In figure 3C the authors choose to focus on *KRAS*, yet there are other genes with a significantly higher difference. The authors should at least mark their names.

We have now highlighted some additional significant differentially expressed genes in the text (lines: 162-166).

2) In figure 4e the relevant text exclaims: “WT S6 contained 62.2% β -like cells compared to 37.7% in KO, whereas S6 KO 187 had still 64.4% PP-like cells compared to 35.5% in WT”. This description is misleading, as it suggests that in KO cells 37.7% are beta, whereas 64.4% are PP-like which is clearly more than 100%. Rather, it should be said: Among the beta cells analyzed, 62.2% were WT and 37.7% KO. Given this, the authors should also present a breakdown of each sample into the cell types which comprise it. This will likely uncover additional cell types which the authors did not mention, and if so – they should present them in a clear fashion.

Thank you for pointing this out. We have now replaced Fig. 4e with a bar plot indicating the percentage of major cell types for S6 WT and *TYK2* KO samples. Additionally, we have now added the percentage of different cell types for S5 and S6 WT and *TYK2* KO samples (Fig. 4f) and their complete breakdown in terms of cell numbers (Supplementary Fig. 6d, e). The corresponding text has been modified accordingly (lines: 196-197).

3) In figure 4K the authors claim that G1 is shorter in *TYK2* KO. However, since they did not show that the overall duration of the cell cycle is unchanged, it might just as well be that S is now longer.

This is an interesting comment. To get further insight of the cell cycle, we have additionally performed the flow cytometry based EdU incorporation in combination with NKX6.1⁺ staining during S5 for WT and *TYK2* KO samples. Indeed, we observed that 21.6% of cells are EdU⁺NKX6.1⁺ in *TYK2* KO samples compared to 13.1% in WT, while the number of EdU⁺NKX6.1⁻ cells is similar in both samples (Fig. 4m, n). This result further strengthens the observation of cell type specific modulation of cell cycle pattern in *TYK2* KO cells at S5 which is crucial for the proper augmentation of NEUROG3 and endocrine commitment. The corresponding text has been modified in the main text file (lines: 213-215).

Reviewer #2 (Remarks to the Author):

The authors studied the role of TYK2 in human induced pluripotent stem cells during beta cell development and response to IFN α using TYK2 ko iPSCs.

The manuscript is well written and the proper controls were used throughout the experiments presented.

We thank the Reviewer for the constructive comments and appreciation. We hope the revision outlined below as per Reviewer's suggestions will further improve the manuscript and clarify our novel findings.

A few comments:

- How come that TYK2 ko had less ins mRNA, less ins content but still respond to glucose stimulation with same levels of ins as WT?

We thank the Reviewer for pointing this out. The dynamic insulin secretion data in the earlier Fig. 2i was presented to show the insulin secreting capacity (kinetics of insulin secretion) of both WT and TYK2 KO S7 mature SC-islets normalized to the secretion at low glucose 2.8mM (y-axis: Fold change INS secretion to G2.8).

To clarify this point we have now also added static insulin secretion data, which is normalized by the total DNA content in the respective SC-islets, showing significantly ($p = 0.006$) reduced absolute insulin secretion in TYK2 KO S7 SC-islets with 30mM KCl stimulation (Fig. 2l) (lines: 123-126).

- Could it be that TYK2 ko is affecting other endocrine cells?

This is a good point. To understand the potential effect of TYK2 KO on the other major endocrine cell types e.g., the α - and δ - cells, we performed immunostaining and quantification for GCG and SST cells in mature S7 WT and TYK2 KO SC-islets. We did not observe any significant differences in either α - or δ - like cell population between WT and TYK2 KO S7 mature SC-islets (Supplementary Fig. 3a, b) (lines: 118-119).

Additionally, stem-cell derived non-pancreatic endocrine cell type, i.e. enterochromaffin-like cells (SC-EC), have been reported with SC-islets differentiation. We have now performed additional immunostaining for the SC-EC marker, SLC18A1 and INS on mature S7 WT and TYK2 KO SC-islets and their quantification (Fig. 2j, k) (lines: 119-122). Interestingly, we do observe the SC-EC cells in our differentiation and importantly, we observed $\approx 31\%$ reduced expression in TYK2 KO SC-islets. Notably, the SC-ECs follow similar *in vitro* development trajectory like β -cells².

- What is the rationale of stimulating progenitor cells with IFN α ?

Thank you for this comment. We chose the immature S6 SC-islets for IFN α treatment to study its effect on all possible early cell types (PP, EP, early β -like cells, α -like cells) with single-cell RNAseq, such a resource is currently unavailable. We hope that you will appreciate the comparative effect of IFN α on all these population e.g., Supplementary Fig. 8a, b. It is relevant for the clinical setting to study the effects of IFN α on the developmentally immature β -cells, since the pathogenesis of T1D has been suggested to initiate very early in the neonatal or even intra-utero period³. Notably, we also performed additional experiments on S7 mature SC-islets with IFN α for all the major observations (Fig. 6; Fig. 7; Supplementary Fig. 9).

- Have you looked HLA-II genes? It would be interesting to see if TYK2 ko also prevents HLA-II upregulation, since it has been recently reported that HLA-II can be expressed in beta cells and could have a role in T1D pathogenesis DOI: 10.1007/s00125-021-05619-9

This is also a very interesting suggestion, thank you. Recent reports³ suggested that HLA- Class II can be overexpressed in β -cells of T1D patients. Our scRNA-seq data show upregulation for the HLA Class

II genes *HLA-DQB1*, *HLA-DRB1* and *HLA-DPB1* in S6 WT SC-islets with IFN α treatment whereas their expression remains mainly unchanged in *TYK2* KO samples (Supplementary Fig. 9a-c). Additionally, we also performed experiments on S7 mature H1 SC-islets with IFN α and +/- *TYK2i*. We observed significant upregulation of *HLA-DQB1* and *HLA-DRB1* but not *HLA-DPB1* with IFN α treatment which was completely prevented in the presence of *TYK2i* (Supplementary Fig. 9d -h) (lines: 256-260; 360-363).

- It has been reported that Neurog3+ was found at higher levels in epsilon cells, how do you explain this in relation to your recent findings? <https://doi.org/10.3389/fendo.2021.736286>

This is an observation from the Wang⁴ group, where they found 10 epithelial like cells out of 11,174 pancreatic cells with higher expression of *NEUROG3* and clustered with Epsilon cells. However, we did not observe any separate epsilon cell type clusters or *NEUROG3* expression in *GHRL* positive cells in our S6 scRNA-seq data of either WT or *TYK2* KO samples. Moreover, in the epsilon cell clusters of our matured S7 SC-islets scRNA-seq data set¹, we also did not observe *NEUROG3* expression.

- Would *TYK2* inhibition prevent also an attack from the autoreactive CD8 T cells already present in the tissue?

This is likely to be the case because the anti-viral T-cell model that we used in the experiments presented (Fig. 7) elicits a much stronger cytotoxicity than the one elicited by autoreactive T cells. Hence, if *TYK2* inhibition is able to reduce these strong anti-viral cytotoxic responses, it is also likely to reduce the weaker autoreactive ones. Importantly, *TYK2* inhibition strongly decreases HLA class I expression (our present data) which would potentially decrease beta cell antigen presentation to CD8 T-cells.

Reviewer #3 (Remarks to the Author):

Chandra et al. studied the role of T1D associated gene TYK2 in β -cell development using the hPSC-based model. They found that TYK2 deletion led to the decreasing of endocrine precursors by regulating KRAS expression. Additionally, they showed that TYK2 knockout or inhibition prevented IFN α -induced antigen presentation and enhanced cell survival against CD8+ T-cell cytotoxicity. In general, this work is interesting, and provide a promising candidate for T1D treatment. However, several key questions need to be addressed:

We are grateful for the positive feedback given by Reviewer 3. We hope the additional data we have collected for the revision will help answer the questions raised.

Major concerns:

1. Based on the conclusions of this work, TYK2i BMS-986165 is a promising T1D drug candidate. It is necessary to test the in vivo function of this small molecule in diabetic mouse model, which will provide broader interests for the field.

We would like to thank the Reviewer for this relevant comment. Indeed, TYK2i BMS-986165 used in this study is already in phase 2-3 clinical trials for psoriasis⁵. Moreover, this compound has also showed positive results in the preclinical models of other immune mediated diseases (lupus nephritis and inflammatory bowel disease⁶).

We agree that it will be important to study the effects of TYK2 inhibition in T1D mouse models, and one of the co-authors of the present study (D.L.Eizirik) has been recently funded by the Juvenile Diabetes Foundation International, in collaboration with Dr Evans-Molina (Indiana University), to perform these experiments. We hope that the Reviewer agrees with us that this large experiment – which will require a couple of extra years of work - is beyond the context of the present manuscript which focuses on human stem-cell derived islets as the main model system.

2. Rescue experiments are required for TYK2 KO.

This is a relevant suggestion. We have now performed additional TYK2 transient overexpression experiments at the key stage S5. The overexpressed TYK2 protein and transcripts were confirmed in the TYK2 KO samples (New, Fig. 3o-q). Importantly, we observed a complete rescue for the KRAS upregulation in TYK2 KO^{OE} samples whereas the effect on NEUROG3 expression was milder in the present experimental condition. These new data have now been added as Fig. 3o-s, and we have updated the main text correspondingly (lines: 175-180).

3. How about the phenotypes of hiPSCs with SNPs (rs34536443 and rs2304256)? Can the authors directly generate hiPSCs from patients with SNPs or generate hiPSCs with these SNPs using CRISPR?

The choice of using TYK2i (BMS-986165) in the present study was also because it targets the pseudokinase domain of TYK2 and allosterically blocks its receptor-stimulated activation, a mechanism analogous to that of the TYK2-deactivating coding variant rs34536443⁶. The TYK2i thus mimics the effect of the SNPs mentioned.

Additionally, we have added phenome-wide association analysis for SNP rs2304256 from the latest R7 dataset of FinnGen project (r7.finnngen.fi), which compiled 3095 clinical endpoints obtained from electronic health record data of 309,154 Finnish individuals. We found that, similar to rs34536443, rs2304256 also provided protection against several autoimmune/auto-inflammatory diseases in the Finnish population, including T1D (8,671 cases and 255,466 controls). This information is now added in Supplementary Fig. 10.

4. How about the phenotypes of overexpression of TYK2?

As explained previously for comment 2, we have overexpressed TYK2 protein in WT S5 cells and assessed the effect on downstream targets. Intriguingly, in contrast to KO cells we did not observe any modulation in *KRAS*. We do not see, however, that a comprehensive analysis of constitutive TYK2 overexpression would add key information to the current manuscript, particularly considering that the TYK2 polymorphisms that lead to protection against autoimmune diseases are all related to decreased and not increased TYK2 activity.

5. The complete loss of TYK2 impaired EP formation thus reduced the number of INS⁺NKX6-1⁺ cells (~ 30% reduction), but why the ratio of INS⁺ cells were comparable in WT and TYK2 KO 2-month grafted tissue sections.

The *in vivo* analysis of graft cells is presented as the proportion of INS⁺ and GCG⁺ cells in WT and TYK2 KO in 2-month grafted tissue sections (y-axis title in the Supplementary Fig. 4e has been now correctly marked). It is difficult to measure the total endocrine volume of the graft for the absolute number of INS⁺ cells. Similar levels of circulating human C-peptide in mice carrying WT and KO grafts (Supplementary Fig. 4c) suggest that TYK2 is no longer needed for the *in vivo* development of functional beta cells.

6. *KRAS* can influence the G1/S transition, only use the expression level of cell cycle markers to estimate the fraction of dividing cells in WT and KO groups is not sufficient. The fraction of G1 and S phase cells should be determined by experiments, such as FACS.

This is indeed an interesting comment. To get further insights on the cell cycle, we have now additionally performed flow cytometry-based EdU incorporation in combination with NKX6.1⁺ staining during S5 for WT and TYK2 KO samples. Indeed, we observed that 21.6% of cells are EdU⁺NKX6-1⁺ in TYK2 KO samples compared to 13.1% in WT, while the number of EdU⁺NKX6-1⁻ cells is similar in both samples (Fig. 4m, n) (lines: 213-215). These data further strengthen the observation of cell type specific modulation of cell cycle pattern in TYK2 KO samples at S5 which is crucial for the proper augmentation of NEUROG3 and endocrine commitment.

Minor concerns:

1. Fig. 2h showed the total insulin content decreased in KO cells. It is better to show the specific insulin content value.

This has now been replaced with the specific insulin content value (Fig. 2h) as suggested.

2. Fig. 3b, although the p-value is significant, the difference between WT and KO samples at S5 is too modest.

The p-value for the PCA is a linear regression comparison of eight values of the PCA vs genotype (ln(PC~genotype)). This information is updated in the figure legend (line 823-824). Notably, we observed the highest numbers of significant upregulated (319) and downregulated (412) genes in S5 samples compared to other stages.

3. The authors showed TYK2 KO could significantly decrease the INS⁺ cells at S6 in Figure 2 and Figure 4, but the INS⁺ cells showed no difference between WT and KO cells in Fig. 6a and b. This should be consistent.

A better representative image is now added (Fig. 6a). The flow cytometry-based quantification for INS is already presented in Fig. 6c, d.

4. In Fig. 4i, j, whether the expression level of CDK2 and CDK4 is significant in PP and EP cells? There should be marked in the figure.

Expression level of CDK4 is significant in both PP and EP cells and this is now indicated in the Fig. 4k whereas CDK2 levels are non-significant.

5. All bioinformatics tools used in this paper have no version information, such as featureCounts, Cutadapt. Detailed information should be provided.

The version information for all the used bioinformatics tools is now added in the Methods section.

6. In line 402 to 404, 'Raw sequencing data...using the Pearson method'. This sentence seems redundant.

This has been now corrected.

7. The criteria for filtering low expression genes using reads count seems unreliable for short or long genes that probabilistically have less or more reads coverage because of gene length. Therefore, it would be recommendable to use normalized gene expression (e.g., FPKM, TPM) for filtering.

It's true that filtering by read coverage is unfavourable for very short transcripts, but we had already eliminated the very short transcripts during library preparation. The size selection of RNA fragments during a standard RNA-seq library construction eliminates fragments below a certain size limit (400bp in our case).

We used only a very mild pre-analysis filtering to get rid of genes that were not expressed or contained mostly unreliably mapped reads (such as reads mapping to low complexity regions like the poly-A tail). If the gene contained more than 50 reads in total, across all the samples in each developmental stage, it was kept in the analysis. About 75% of the genes that were filtered out had zero mapped reads (i.e., were not expressed). These would be eliminated regardless of the filtering strategy. The DESeq2-package that was used for the differential expression analysis uses read count data and does gene filtering during the analysis based on read counts anyway. Adding an additional FPKM or similar filtering step would not help for the very low or non-expressed genes. In the attached figure (for the Reviewer only) you can see the distribution of FPKM values vs read counts (both log_{1p}-transformed). All the filtered genes have very low FPKM values. If we extended the filtering using the highest FPKM value of the filtered genes we would eliminate a large number of reliably mapped genes.

8. In line 435, for definition of differentially expressed genes, the author didn't note the threshold for expression fold change. And the DEGs showed in Fig. 3c seems have even lower p-values but not 0.01.

We didn't use any arbitrary fold change cut-off to define differential expression. We only used false discovery rate corrected *p*-values (FDR<0.01).

9. In line 156-158, the author indicated that NEUROG3 and NKX2-2 were significantly down-regulated at S5. However, the author didn't show the significance test (e.g., Expression fold change and FDR calculated by DESeq2).

The *p*-value which is calculated with DESeq2 now indicated in the figure (Fig. 3d, e) and the corresponding figure legend text is also updated (line 828).

10. In Figure 3f, the author showed the results of Reactome enrichment analysis but titled with GO enrichment. Additionally, x axis should note with percentage.

Thank you for pointing this out: we have now corrected the title in the figure (Fig 3f). Now, x-axis is represented with -percentage of DE genes in category / all genes in category (%).

References:

1. Balboa, D., Barsby, T. & Lithovius, V. Functional, metabolic and transcriptional maturation of human pancreatic islets derived from stem cells. doi:10.1038/s41587-022-01219-z.
2. Veres, A. *et al.* Charting cellular identity during human in vitro β -cell differentiation. *Nature* doi:10.1038/s41586-019-1168-5.
3. Rewers, M. *et al.* PATHOGENESIS OF TYPE 1 DIABETES (A PUGLIESE AND SJ RICHARDSON, SECTION EDITORS) The Environmental Determinants of Diabetes in the Young (TEDDY) Study: 2018 Update. doi:10.1007/s11892-018-1113-2.
4. Yong, H. J. *et al.* Gene Signatures of NEUROGENIN3+ Endocrine Progenitor Cells in the Human Pancreas. *Frontiers in Endocrinology* **12**, (2021).
5. Papp, K. *et al.* Phase 2 Trial of Selective Tyrosine Kinase 2 Inhibition in Psoriasis. *New England Journal of Medicine* **379**, 1313–1321 (2018).
6. Burke, J. R. *et al.* Autoimmune pathways in mice and humans are blocked by pharmacological stabilization of the TYK2 pseudokinase domain. *Sci. Transl. Med* vol. 11 (2019).

REVIEWERS' COMMENTS

Reviewer #1 (Remarks to the Author):

The reviewers had answered all of the raised issues, but two minor matters still require attention:

1) In Figure 2K, the authors indicate reduction in the number of slc+ cells following Tyk2 KO and present bar plots which indicate measurement of slc+ cells over several repetitions. However, the nature of these repetitions should be clarified - is it counting different clusters in one batch of differentiation, or did they actually repeat the experiment 13 times? How many cells were counted?

2) Although the authors now address the presence of enterochromafin-like cells, it is unclear why these cells are absent from scRNAseq analysis.

After addressing these issues correctly, my recommendation is that this interesting and thorough manuscript should be accepted for publication.

Reviewer #2 (Remarks to the Author):

Reviewer #3 (Remarks to the Author):

The authors have revised their manuscript extensively and provided essential experimental results and reasonable explanations. Overall, it is an interesting paper for the field, and I would recommend it for publication.

REVIEWERS' COMMENTS

Reviewer #1 (Remarks to the Author):

The reviewers had answered all of the raised issues, but two minor matters still require attention: 1) In Figure 2K, the authors indicate reduction in the number of slc⁺ cells following Tyk2 KO and present bar plots which indicate measurement of slc⁺ cells over several repetitions. However, the nature of these repetitions should be clarified - is it counting different clusters in one batch of differentiation, or did they actually repeat the experiment 13 times? How many cells were counted?

We would like to thank Reviewer #1 for their constructive feedback on our manuscript, as well as their appreciation for the new experiments carried out to answer all the raised questions.

Thank you for pointing out this missing information. The SLC18A1⁺ cells were quantified from two biologically different experiments with 5-7 images per experiment. Each image quantified was plotted on the bar plot. The total number of cells quantified for WT and KO S7 cells were 10516 and 11990 cells, respectively. We have now added this information in the legend of Fig. 2k.

2) Although the authors now address the presence of enterochromaffin-like cells, it is unclear why these cells are absent from scRNAseq analysis.

Thank you for this comment. Kindly note that we do observe enterochromaffin-like cells as a subset of the early β -like cells cluster in the scRNAseq analysis, which are positive for *TPHI/LMX1A/SLC18A1* markers. Importantly, the number of cells expressing these markers are reduced in the KO compared to WT in both S5 and S6 samples (Figure R1), in line with the further observation of reduced number of SLC18A1⁺ SC-EC like cells at S7 (Fig. 2k).

Figure R1: scRNAseq based analysis showing the percentage of cells expressing *TPHI*, *LMX1A* and *SLC18A1* in S5 and S6 WT and *TYK2* KO samples.

After addressing these issues correctly, my recommendation is that this interesting and thorough manuscript should be accepted for publication.

We sincerely thank you for all the valuable comments and recommendation.

Reviewer #2 (Remarks to the Author):

We hope that we have answered all the raised concerns from Reviewer #2, and would like to once again thank you for the previous constructive comments and feedback.

Reviewer #3 (Remarks to the Author):

The authors have revised their manuscript extensively and provided essential experimental results and reasonable explanations. Overall, it is an interesting paper for the field, and I would recommend it for publication.

We are genuinely happy and thankful to have answered the relevant comments from Reviewer #3 and immensely appreciate the constructive feedback, recognition of the revised work and the valuable recommendation.